# Hybrid Architectures Used in the Protection of Large Healthcare Records Based on Cloud and Blockchain Integration: A Review

**Leonardo Juan Ramirez Lopez \*** , **David Millan Mayorga, Luis Hernando Martinez Poveda, Andres Felipe Carbonell Amaya and Wilson Rojas Reales**

Osiris & Bioaxis Research Group, Engineering Faculty, Universidad El Bosque, Bogota 111321, Colombia; lmartinezpov@unbosque.edu.co (L.H.M.P.); acarbonell@unbosque.edu.co (A.F.C.A.)
\* Correspondence: ljramirezl@unbosque.edu.co; Tel.: +57-3114905014

**Abstract:** The management of large medical files poses a critical challenge in the health sector, with conventional systems facing deficiencies in security, scalability, and efficiency. Blockchain ensures the immutability and traceability of medical records, while the cloud allows scalable and efficient storage. Together, they can transform the data management of electronic health record applications. The method used was the Preferred Reporting Items for Systematic Reviews and Meta-Analyses (PRISMA) methodology to choose and select the relevant studies that contribute to this research, with special emphasis set on maintaining the integrity and security of the blockchain while tackling the potential and efficiency of cloud infrastructures. The study's focus is to provide a comprehensive and insightful examination of the modern landscape concerning the integration of blockchain and cloud advances, highlighting the current challenges and building a solid foundation for future development. Furthermore, it is very important to increase the integration of blockchain security with the dynamic potential of cloud computing while guaranteeing information integrity and security remain uncompromised. In conclusion, this paper serves as an important resource for analysts, specialists, and partners looking to delve into and develop the integration of blockchain and cloud innovations.

**Keywords:** blockchain; cloud; security; healthcare; integrity



## 1. Introduction

In our increasingly digital world, the integration of blockchain technology and cloud computing has emerged as a promising innovation to address the challenges of data storage. As the volume of data continues to grow exponentially, securely storing, managing, and retrieving large files has become a critical issue. Blockchain technology, renowned for its decentralized and immutable ledger, provides a robust framework for securing sensitive health information. By encrypting data and recording transactions in a tamper-proof manner, blockchain ensures data integrity, authenticity, and auditability, thereby mitigating the risk of unauthorized access, fraud, and data manipulation [1,2]. Cloud storage, on the other hand, offers unparalleled scalability and accessibility, allowing healthcare organizations to store, process, and analyze vast amounts of data with ease. Cloud-based solutions facilitate seamless data sharing and collaboration among healthcare stakeholders, enabling real-time access to critical information and fostering innovation in care delivery [3]. However, concerns around data security and privacy persist due to the centralized nature of cloud storage [4].

Combining the security of blockchain with the practical data storage capacity of the cloud represents an opportunity [4]. This study aims to explore the integration of blockchain and cloud computing in healthcare data management, with a focus on addressing security, scalability, and interoperability challenges. By synthesizing existing research and identifying key gaps, this study contributes to enhancing our understanding of the

potential benefits and limitations of this integration [4–19]. We thoroughly examine the pertinent literature, drawing from numerous scholarly contributions on this topic. Our exploration covers the technical dimensions of blockchain, including its service models and security frameworks, along with analyzing the performance implications when integrated with cloud data centers [20,21].

The integration of blockchain and cloud computing in healthcare data management holds particular significance. Healthcare data, characterized by its sensitive nature, requires secure and scalable management solutions [22]. Integrating blockchain's immutability and cryptographic security with the scalability and cost-effectiveness of cloud computing has the potential to revolutionize healthcare data management [4,8,23]. By ensuring data integrity, enhancing security, and enabling interoperability, this integration can facilitate efficient and transparent healthcare data exchange, leading to improved patient care and outcomes. Additionally, the transparent and auditable nature of blockchain transactions facilitates data sharing for research purposes while preserving patient privacy and confidentiality [7]. Leveraging cloud-based analytics and machine learning algorithms enables healthcare stakeholders to derive actionable insights from vast datasets, driving personalized medicine, predictive analytics, and population health management initiatives.

### 1.1. Discussion of Challenges and Limitations in Field of Study

- **Efficiency and Scalability:** Transaction efficiency and scalability are essential issues in the blockchain. Adding interactions with the cloud can increase complexity and the time required to complete transactions, which can reduce efficiency. Scalability becomes a problem when handling large volumes of data from the cloud to the blockchain, leading to network congestion and longer processing times [24,25].
- **Costs and Sustainability:** Implementing blockchain and cloud solutions can be costly, both in terms of infrastructure and energy consumption. Sustainability and energy efficiency are important considerations, especially at a time when the carbon footprint of blockchain is under scrutiny. Researching more energy-efficient solutions is crucial [26].
- **Interoperability and Standards:** Interoperability between different blockchains and cloud providers can be a problem. The lack of common standards and protocols can hinder seamless integration. Researching and developing standards to ensure a smooth interaction is essential [25].
- **Quantum Computing Resistance:** With the development of quantum computing, resistance to quantum attacks becomes a concern. Blockchain and cloud must be resilient to these emerging technological challenges [27].
- **Centralization vs. Decentralization:** Striking the balance between centralization and decentralization is a dilemma. Some solutions may require a degree of centralization, which goes against the fundamental principles of blockchain. Finding the right balance is essential [28–30].

### 1.2. Study Objectives and Motivation behind Them

The study aims to examine and provide a comprehensive review of research that integrates cloud and blockchain technologies. This is with the objective of exploring and learning the existing ways of accomplishing a successful integration between these two technologies and large healthcare files. Therefore, to achieve this, the Preferred Reporting Items for Systematic Reviews and Meta-Analyses, from now on referred to as PRISMA methodology, is used to explore and sort the papers among the databases. For this reason, the following databases were selected among others due to the relevance and amount of content that can be found in them. Scopus, Engineering Village, MDPI, Elsevier, Google Scholar, and IEEE were the selected databases from which the papers were extracted for this review. As a result of this overarching objective, the investigation to address specific research questions (RQ) is as follows:

RQ1: What is the distribution of papers across different years?

RQ2: How are the chosen papers related to the proposed keywords?
RQ3: Which of the papers explores blockchain and cloud computing as a review?
RQ4: Which papers explore blockchain–cloud computing and healthcare?
RQ5: In the current landscape of secure blockchain and cloud integration, what constitutes the primary challenges that organizations and practitioners face?
RQ6: How are large healthcare files currently secured in blockchain–cloud integration, and what methods and techniques are currently being utilized?

*1.3. Contributions*

Throughout this study, we conduct a comprehensive review of papers related to blockchain, cloud, security, and healthcare. This involved carefully selecting papers that met specific parameters established by us, from among thousands available. Our criteria for paper selection dictated that only papers released between 2017 and 2023 were considered, which aimed at avoiding outdated architectures or methodologies.

The study aims to provide an extensive overview of the secure integration of blockchain technology and cloud computing in the contemporary landscape. The primary contributions of our research can be summarized as follows:

- Synthesis of key findings to provide a comprehensive understanding of the current state of secure integration in the given domains. Classification based on topics and keywords.
- Recommendations for future research and potential areas for improvement in the integration of blockchain technology and cloud computing for enhanced security in healthcare applications.
- Analysis of emerging trends and innovative approaches in the secure integration of blockchain and cloud computing.
- Exploration of challenges and field limitations.

**2. Material and Methods**

Extensive prior work exists exploring blockchain–cloud integration with an emphasis on security and chain integrity has been extensively explored with a hybrid blockchain architecture for Cloud Manufacturing-as-a-Service (CMaaS) platforms, emphasizing improved data storage efficiency. As mentioned previously, a hybrid blockchain architecture for Cloud Manufacturing-as-a-Service platforms could help to improve data storage efficiency. This seminal work demonstrated the advantages of a hybrid blockchain approach for enhanced data management [25].

Likewise, another paper explores regulatory compliance in multi-cloud blockchain deployment. By decentralizing identity, their user-centric model aimed to follow GDPR while exploiting blockchain's security properties across interconnected clouds. Their insights advanced compliance along with unlocking blockchain's potential [31,32].

Healthcare data integrity motivated new techniques like those proposed in [26], which proposed a blockchain-enabled bioacoustics signal authentication system for cloud-based electronic medical records. This work emphasized the importance of securing medical data through innovative authentication techniques.

Moreover, the reviews presented in papers [2,33] provided a comprehensive overview of blockchain technology in healthcare, emphasizing its role in ensuring data security and privacy. Their authoritative overview illuminated blockchain's prospects and challenges, cementing foundational knowledge.

Looking beyond healthcare, [20] explored the role of blockchains and decentralized oracle networks in technology-enabled financing for sustainable infrastructure. Their focus on secure and decentralized financial transactions within a blockchain context contributed to understanding the broader applications beyond healthcare.

Additionally, studies such as Berdik D. et al. [34] and Taghavi et al. [35] delved into secure access frameworks and reliability models for blockchain oracles, respectively, shedding light on crucial security aspects in blockchain-enabled systems.

Collectively, this extensive research reinforces the rising significance of security and chain integrity in multidisciplinary blockchain–cloud integration, with far-ranging potential from healthcare to finance.

### 2.1. PRISMA

PRISMA is a protocol that consists in evidence-based minimum set of items for reporting in systematic reviews and meta-analyses. It primarily focuses on the reporting of reviews evaluating the effects of interventions but can also be used as a basis for reporting systematic reviews with objectives other than evaluating interventions (e.g., evaluating etiology, prevalence, diagnosis, or prognosis) [36]. It makes understanding what needs to be done at each stage of the review easier. This protocol helps to refine the research question further, serving as a foundational tool for systematic reviews across various fields.

Why PRISMA?

PRISMA is a valuable tool in research due to its role in enhancing the transparency, reliability, and overall quality of systematic reviews and meta-analyses. By providing a standardized reporting framework, PRISMA ensures that researchers clearly articulate their methods, from the systematic search process to study selection and data extraction. Also, it offers several benefits in online training for students, educators, researchers, and readers:

- Reduction of bias: PRISMA includes guidelines that aim to reduce bias in systematic reviews. Transparent reporting helps readers assess the risk of bias in the included studies, leading to a more accurate interpretation of the evidence.
- It enables self-regulated learning by providing systematic search procedures (identification, screening, eligibility, inclusion) via online platforms.
- It serves as a valuable guide for postgraduate students and researchers in conducting comprehensive searches to find necessary papers.
- It aids readers by offering a clear understanding of the process, enabling easy tracking of information sources through systematic review records, and simplifying the evaluation of reported systematic reviews.
- Support for evidence-based practice: PRISMA contributes to the production of high-quality evidence that can be used to inform evidence-based practice, clinical guidelines, and policy decisions.

### 2.2. Statistics of PRISMA Use in Scientific Review Type Articles

There is an escalating interest in meta-analyses within the realm of the biomedical literature. A PubMed search, confined solely to articles classified (by publication type) as meta-analyses, reveals an exponential surge in the annual count of published meta-analyses within this bibliographic database. This count escalated from a solitary publication in 1982 to a staggering 5426 in 2012. It is reasonable to infer that a comparable escalation has transpired across other bibliographic databases, and this upward trajectory is anticipated to persist in the forthcoming years. Pertaining to systematic reviews, a recent scholarly article approximates that an average of eleven such reviews are disseminated daily [37].

The PRISMA guidelines have garnered extensive support and implementation, as demonstrated by their joint publication across numerous journals, references in over 60,000 reports (Scopus, August 2020), approval from nearly 200 journals and systematic review entities, and utilization across a range of disciplines. Observational studies indicate that employing the PRISMA 2009 statement correlates with a more comprehensive reporting of systematic reviews, though there is room for enhancing compliance with the guideline [38].

### 2.3. Importance of PRISMA

The PRISMA statement, originally published in 2009, was made to help systematic reviewers clearly describe why the review was done, how the research was carried out, what the authors did, and what they found. The PRISMA 2020 statement replaces the

2009 statement and includes new reporting guidance that reflects advances in methods to identify, select, appraise, and synthesize studies [38].

Systematic reviews fulfill numerous essential functions. They can offer a consolidation of the existing knowledge in a particular field, from which future research priorities can be identified; they can address questions that otherwise could not be answered by individual studies; they can identify issues in primary research that need to be corrected in subsequent studies; and they can formulate or assess theories regarding the occurrence or explanation of phenomena [38].

PRISMA can be considered a standard in research; it has become a widely accepted reference in the scientific community and has been adopted by numerous journals and organizations as a standard for presenting systematic reviews.

### 2.4. Search Engines and Search Equations

Toward the selection of the papers, various criteria and filters established by us were considered, aiming to separate those papers that could contain outdated information or focus on other areas of study. The first step was to formulate the search equations, which focused on the proposed keywords that are mostly related to our research questions:

- "Blockchain and Cloud Storage Integration" OR "Blockchain and Cloud Computing" OR "Secure Cloud Storage with Blockchain" OR "Decentralized Cloud Storage" OR "Blockchain-Based Data Security in Cloud" OR "Blockchain for Large File Storage" OR "Blockchain and Cloud Security" OR "Cloud-Based Blockchain Applications" OR "Blockchain for Data Integrity in Cloud" OR "Cloud Exchange with Blockchain" OR "Blockchain and Supply Chain Management" OR "Blockchain in Operations and Supply Chain" OR "Blockchain and Cloud Services" OR "Blockchain in Cloud Infrastructure" OR "Blockchain-Based Cloud Services" OR "Blockchain-Based Cloud Storage Solutions".
- Blockchain AND Security AND Cloud
- ("All Metadata": Blockchain) AND ("All Metadata": Cloud)
- Blockchain and cloud and security and Healthcare
- Blockchain AND oracles

The search engines used were Elsevier (ScienceDirect), Google Scholar, IEEE Xplore, and MDPI. As mentioned before, only papers from 2017 onwards were selected; in the first stage, we encountered a massive number of papers, above 23 thousand, adding up all the four selected databases. As a result of this huge number of papers, we established criteria to separate those papers that moved away from our purpose and objective. After this first-year-of-publish filter, we decided to remove all those scripts that talked about cryptocurrencies or any other topic but secure blockchain applied to cloud computing. After applying some other filters (Figure 1) (for further information about PRISMA check Appendix A, where the PRISMA for each database will be found), we ended up having over 100 papers that served our purpose.

### 2.5. Incorporation and Exclusion Parameters

Studies incorporated in our comprehensive review should suggest an investigation around a secure integration of blockchain and cloud computing or suggest viable ways to connect a chain of blocks with data stored in the cloud. Also, we consider those papers that mention cloud, blockchain, and healthcare. Table 1 provides more details about the inclusion/exclusion criteria used in the papers.

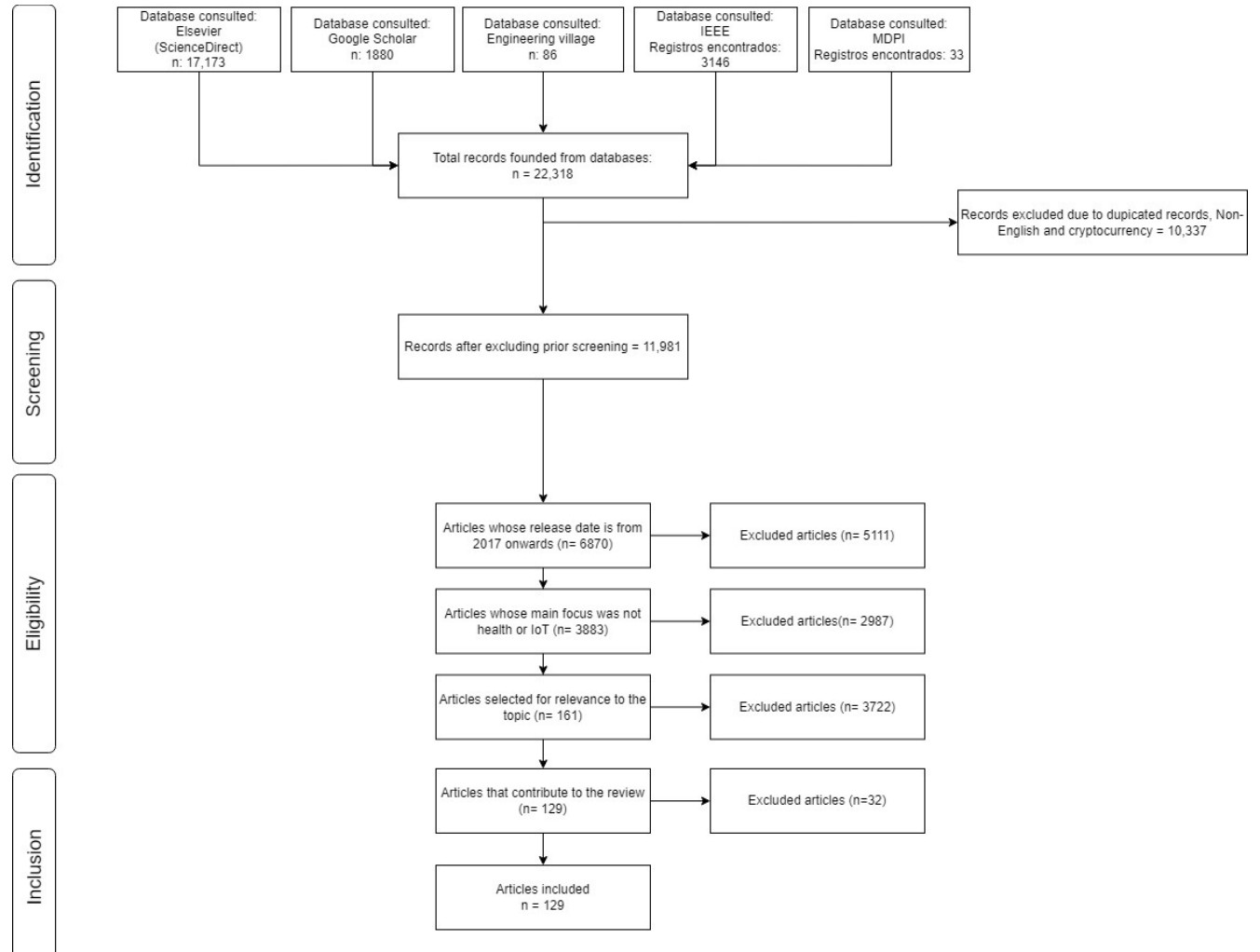

**Figure 1.** Flowchart for selected papers based on PRISMA.

**Table 1.** Criteria of inclusion/exclusion.

| Papers Included | Papers Excluded |
| --- | --- |
| Papers must talk about blockchain and cloud integration. | Papers that focus on cryptocurrencies or its focus is Internet of Things. |
| Papers that mention ways of securely connecting the cloud with blockchain and show the process. | Papers published before 2017. |
| Review papers that help the purpose and objective of this paper. | Papers that proposed storing the chain in the cloud. |

## 3. Results

In this section, we present the outcomes of our comprehensive review focused on the integration of blockchain and cloud technologies within the healthcare domain. Our analysis, involving an exhaustive examination of over 100 selected papers, aimed to discern patterns, trends, and insights pertaining to the current landscape of blockchain–cloud secure integration in managing large healthcare files.

The results presented herein are encapsulated through a series of tables and graphics meticulously crafted by our team. These visual representations serve to categorize and dissect the diverse range of papers we scrutinized, shedding light on key themes, methodologies, and emerging areas of interest within the intersection of blockchain, cloud, and healthcare.

### 3.1. Landscape of Healthcare-Related Blockchain Applications for Security and Privacy

Maintaining the security and privacy of patient data is of utmost importance in the healthcare field. To safeguard patients' rights and prevent unauthorized individuals from accessing sensitive information, it is essential to implement robust security measures for electronic health records (EHRs). The integration of blockchain technology is viewed as a promising approach to bolster the security and confidentiality of EHRs. By utilizing blockchain, EHR systems can provide a secure and immutable platform for storing and sharing patient data, thereby minimizing the risk of data breaches (Table 2) [39].

The decentralized architecture of blockchain technology serves as a robust deterrent against data breaches and maintains the integrity of sensitive information. Incorporating smart contracts and cryptocurrency transactions adds an additional layer of fortification, rigorously restricting access to only those users with proper authorization. A particularly compelling aspect is the patient-centric control over access permissions, empowering individuals to single-handedly dictate which entities can view their personal data and dynamically modify these permissions as they deem necessary. Leveraging blockchain for access control grants patients the ability to bestow or rescind data access privileges at any juncture, thereby ensuring that confidential information remains exclusively accessible to authorized parties, consequently bolstering the sanctity of data privacy [40,41].

**Table 2.** Landscape of blockchain technology and healthcare.

| Work | Title | Content |
|---|---|---|
| [42] | A Privacy Preserving Framework for Health Records using Blockchain | A blockchain-based privacy-preserving framework for secure storage and transfer of electronic health records (EHR). Doctors upload the EHR which is encrypted using the SHA256 hashing algorithm and stored as separate blocks on the blockchain. Patients have complete control over their EHR and can share their health records with doctors at various medical institutions through a unique key shared via the doctor's email. Block validation is done using the Delegated Proof-of-Stake (DPoS) consensus algorithm, which guarantees the privacy of the patient's data. |
| [43] | Privacy Preservation and Access Control for Sharing Electronic Health Records Using Blockchain Technology | It presents a secure blockchain solution with smart contracts to enable the privacy-preserving sharing of electronic medical records between patients and providers while managing granular access controls. The performance scales reasonably with larger file sizes. |
| [44] | Scalable blockchain model using off-chain IPFS storage for healthcare data security and privacy | Proposes a decentralized framework that integrates blockchain technology with the Interplanetary File System (IPFS) for secure and patient-centric management of electronic health records (EHR). The blockchain provides an immutable and tamper-proof distributed ledger, while the IPFS enables the off-chain storage of encrypted EHR files to overcome blockchain scalability limitations. The framework employs a patient-centric access model where patients control the sharing of their encrypted health data stored on IPFS with healthcare providers. |
| [45] | Blockchain-Based Access Control Model to Preserve Privacy for Personal Health Record Systems | The article proposes a blockchain-based personal health record (PHR) system model that aims to address several drawbacks of using blockchain for PHR systems, such as limited storage, privacy concerns, irrevocable consent, inefficient performance, and high energy consumption. The proposed model leverages blockchain's immutability and tamper resistance features while employing proxy re-encryption and other cryptographic techniques to preserve data privacy. Key features of the model include fine-grained and flexible access control, the revocability of consent, auditability, and tamper resistance. The PHR data are encrypted and stored on cloud storage for availability, while metadata are stored on a private blockchain for tamper resistance. |

The articles propose various blockchain-based frameworks and models to enable secure, privacy-preserving storage and the sharing of electronic/personal health records while overcoming blockchain's limitations through integration with other technologies like IPFS, cloud storage, and cryptographic primitives like encryption and access control mechanisms.

Another blockchain-based privacy-preserving system has doctors uploading encrypted EHRs that are stored as separate blocks, with patients controlling the sharing of keys for record access across institutions. It employs the Delegated Proof-of-Stake consensus for privacy-preserving block validation, reducing resource needs, computational costs, and transaction times compared to traditional approaches. Some frameworks leverage Ethereum's smart contracts to manage relationships and access permissions between EHR owners (patients) and users (providers), scaling reasonably for larger file sizes.

To address blockchain's drawbacks for personal health records (PHRs) like limited storage and privacy concerns, one proposed model integrates blockchain for tamper resistance, proxy re-encryption for data privacy, and cloud storage for availability. It provides fine-grained access control, revocable user consent, auditability, and tamper-proof PHR storage. Encrypted PHR data reside in the cloud while metadata are maintained immutably on a private blockchain. Comprehensive security analyses prove the models' ability to preserve privacy and prevent tampering, with performance improvements over existing PHR systems.

### 3.2. Results Based on the Proposed Research Questions

**RQ1:** What is the distribution of papers across different years?

As shown in Table 2, there are no papers included before 2017 due to the lack of papers that talk about blockchain and cloud integration. Besides the lack of papers, those few papers that can be found before 2017 could have outdated data or propose solutions that nowadays are not viable because they can become obsolete. The data reveals a progressive increase in scholarly contributions over time, reaching a peak in 2022 with 23 papers. The surge in publications from 2020 onwards suggests a growing interest and heightened focus on the integration of blockchain and cloud technologies within the context of healthcare. This temporal trend underscores the contemporary relevance and evolving nature of research in this domain, showcasing the increasing importance and recognition of blockchain–cloud integration in managing large healthcare files.

**RQ2:** How are the chosen papers related to the proposed keywords?

During the research for papers, we encountered that most of the good papers were trying to go beyond a simple connection between blockchain and cloud; they were exploring options not only to make a connection but to make it secure while keeping the integrity of the chain. As shown in Table 3, most of the papers talk about security and blockchain–cloud or chain integrity. In this sense, Table 4 shows the classification of articles based on keywords. On the other hand, Figure 2 shows, in percentage form, how the topics of the papers are distributed; based on the graph we can say that most of them speak of blockchain–cloud security.

**Table 3.** Distribution of papers by year.

| Year | Number of Papers |
| --- | --- |
| 2017 | 6 |
| 2018 | 6 |
| 2019 | 12 |
| 2020 | 26 |
| 2021 | 21 |
| 2022 | 29 |
| 2023 | 29 |

**Table 4.** Classification of research papers on blockchain and cloud based on given keywords.

| Work | Security | Review/ Survey | Blockchain–Cloud | Chain Integrity | Healthcare |
|---|---|---|---|---|---|
| [46,47] | X | X | X | X | |
| [15,26,28,31,46–65] | X | | X | X | |
| [4,5,10–14,16,18–21,23–25,27,32,35,66–91] | | | X | X | |
| [49,92,93] | X | | X | | |
| [3,30,94–96] | | | X | | |
| [1,6,7,9,17,29,97–99] | | X | X | X | |
| [100] | X | | | | |
| [10,65] | | | X | X | X |

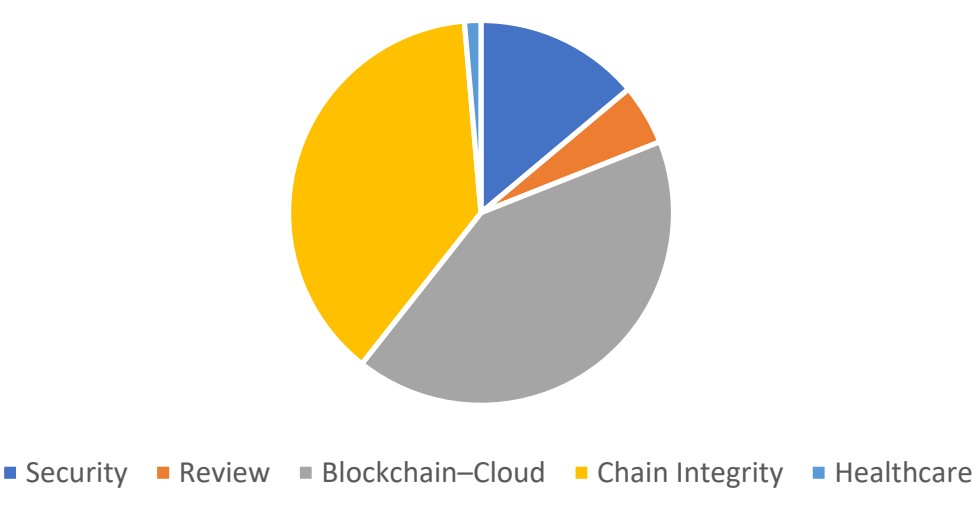

Distribution of different fields

- Security
- Review
- Blockchain–Cloud
- Chain Integrity
- Healthcare

**Figure 2.** Application fields of the papers.

**RQ3:** Which of the papers explores blockchain and cloud computing as a review?

Figure 2 shows the percentage that represents the application fields of the papers. It should be noted that in the graph the lowest percentage is in those papers that include healthcare. Nonetheless, we were able to find four papers that were reviews and had a similar focus in their research.

**RQ4:** Which papers explore blockchain, cloud computing, and healthcare, and what do they contribute?

The presented papers enlighten the challenges, intricacies, and prospective ways for exploration in the context of integrating blockchain and cloud technologies within the healthcare domain. These scholarly works provide comprehensive insights into the intricate dynamics, outlining the nuanced difficulties associated with such integration. Additionally, they articulate potential trajectories for navigating the complex landscape of merging blockchain and cloud in healthcare settings (Table 5).

**Table 5.** Papers that explore blockchain and cloud computing as a review.

| Work | Title | Area of Focus | Content |
|---|---|---|---|
| [99] | Integration of Block-chain and Cloud Computing: A Review. | Blockchain, Cloud Computing, and security. | Explores the rising use of blockchain for enhancing cloud data security across various sectors. Suggests integrating blockchain to address vulnerabilities in centralized cloud computing systems. The focus is on reviewing the benefits and applications of cloud-based blockchain services, emphasizing current trends and security challenges. |
| [1] | Blockchain and Cloud Computing-A Review | Blockchain and Cloud Computing | Examine the literature on blockchain-based enterprise solutions from 2008 to 2021. It explores three categories: blockchain using IaaS, PaaS, and SaaS, discussing characteristics and their relation to cloud services. The study investigates cutting-edge applications in ledger storage, strategy creation, computation, data aggregation, micro-services, and extraction. The report concludes with current issues, expected obstacles, and potential opportunities in blockchain-based cloud technology, aiming to contribute to a comprehensive understanding and the future development of cloud computing environments. |
| [8] | Literature Review of Blockchain-based Cloud Computing: Data Security Issues and Challenges | Blockchain, Cloud Computing, Security Issues, and Challenges | This paper highlights the growing acceptance of cloud computing for handling IT infrastructure and data services efficiently. Also, it explores how blockchain technology, known for its incorruptible nature, can address security issues in cloud applications. As well, the paper emphasizes the importance of security in realizing the benefits of both cloud computing and blockchain. It proposes a literature review to examine how academics utilize blockchain to enhance cloud data security. |
| [99] | Blockchain Technology Application in Security: A Systematic Review | Blockchain, Cloud Com-putting, and security. | The study focuses on categorizing blockchain types, consensus mechanisms, smart contract usage, and integration with other software-based algorithms. The authors emphasize the increasing popularity of blockchain beyond digital currencies, particularly in securing networks. The systematic review identifies the Internet of Things (IoT) as the primary field where blockchain enhances security. |

**RQ5:** In the current landscape of secure blockchain and cloud integration, what constitutes the primary challenges that organizations and practitioners face?

As previously mentioned, there are several challenges to accomplish a secure integration between blockchain and cloud; that is why the papers that will come up next are the most valuable for the purpose of this research. Within Table 6, the most relevant challenges will be shown for each paper mentioned in Table 6. Farther, Table 7 show the main challenges in the current landscape of secure blockchain and cloud integration.

**Table 6.** Analysis of blockchain and cloud integration.

| Work | Title | Contribution(s) |
|---|---|---|
| [2] | A Critical Analysis of Progress and Challenges in the Last Five Years | The paper significantly contributes by objectively evaluating the impact of blockchain technology in the healthcare sector, drawing insights from a thorough analysis of 124 papers published by MDPI over the past five years. Its noteworthy identification of advancements, such as improved data security and interoperability, adds depth to our understanding of blockchain's positive influence on healthcare. |
| [33] | A. Modernizing the Legacy Healthcare System to Decentralize Platform Using Blockchain Technology. | The authors aim to address challenges related to complex medical procedures, large-scale medical data management, and cost optimization. The paper reviews the existing literature and proposes workflows for better data management, implemented using the Ethereum blockchain platform. The feasibility of the proposed system is analyzed in terms of associated costs, and a model-driven engineering approach is used to recover the architecture of traditional healthcare systems. |
| [101] | Design of Secure Protocol for Cloud-Assisted Electronic Health Record System Using Blockchain | This paper addresses the challenges of electronic health record (EHR) management in traditional systems and proposes a secure protocol using blockchain and cloud computing. The authors highlight the potential of blockchain technology to enable the sharing of EHRs across various medical service centers, promoting decentralization and data integrity. However, the integration of cloud computing into the EHR system introduces security vulnerabilities, as sensitive data are transmitted over public channels. The proposed secure protocol aims to address these challenges by using blockchain for data integrity and access control, while the cloud server manages and stores patient EHRs securely. Elliptic curve cryptosystems (ECC) are employed for secure health data sharing within the cloud computing environment. |

**Table 7.** Main challenges in the current landscape of secure blockchain and cloud integration.

| Work | Title | Challenges |
|---|---|---|
| [8] | Literature Review of Blockchain-based Cloud Computing: Data Security Issues and Challenges | Data privacy: Blockchain-based cloud computing presents challenges in ensuring data privacy, as the data are stored in a decentralized manner and is accessible to all nodes in the network. Data integrity: Ensuring data integrity is a challenge in blockchain-based cloud computing, as the data are stored in a decentralized manner and is accessible to all nodes in the network. Scalability: Blockchain-based cloud computing presents scalability challenges; as the number of nodes in the network increases, the time required to reach consensus increases. Interoperability: Interoperability is a challenge in blockchain-based cloud computing, as different blockchains may have different protocols and standards. Regulatory compliance: Blockchain-based cloud computing presents regulatory compliance challenges, as the regulatory framework for blockchain technology is still evolving. |
| [29] | Integrated Blockchain and Cloud Computing Systems: A Systematic Survey, Solutions, and Challenges | Cloud computing introduces new security challenges in secure service management and control, privacy protection, data integrity protection in distributed databases, data backup, and synchronization. Blockchain can be leveraged to address these challenges, partly due to the underlying characteristics such as transparency, traceability, decentralization, security, immutability, and automation. Also, the team explores how cloud computing can affect blockchain, especially the performance improvements that cloud computing can provide for the blockchain. |

**Table 7.** *Cont.*

| Work | Title | Challenges |
|---|---|---|
| [33] | A. Modernizing the Legacy Healthcare System to Decentralize Platform Using Blockchain Technology | Migrated classes: Ensure that the migrated classes are compatible with the blockchain platform. This requires a deep understanding of the blockchain architecture, and the programming languages used to write smart contracts.<br>Patient mobility: When patients move from one hospital to another, their data may be dispersed among multiple hospitals, making it difficult for them to access their medical records. |
| [95] | Toward Decentralized Cloud Storage With IPFS: Opportunities, Challenges, and Future Considerations | Content availability: IPFS relies on peers to host content, which can lead to content unavailability if the peers hosting the content go offline.<br>Content discovery: IPFS uses content addressing to locate content, which can be challenging when the content is not popular or has not been accessed recently.<br>Content integrity: IPFS does not provide any guarantees about the integrity of the content, which can be compromised if the content is modified by a malicious peer.<br>Content privacy: IPFS does not provide any privacy guarantees, which can lead to privacy violations if the content is accessed by unauthorized parties.<br>Content distribution: IPFS does not provide any mechanisms for incentivizing peers to host content, which can lead to the uneven distribution of content. |
| [102] | Comprehensive review for healthcare data quality challenges in blockchain technology | It highlights blockchain's inherent features such as decentralized storage, distributed ledger, immutability, security, and authentication, which have facilitated its practical adoption across various industries, including healthcare. The study analyzes 65 articles from 2016 onwards to identify data quality issues in healthcare blockchain adoption, categorizing these challenges into three domains: adoption, operational, and technological. Despite blockchain's potential to enhance transparency, traceability, privacy, and security, it faces significant challenges such as integration with legacy systems, protection of sensitive data, and regulatory compliance. The review aims to support professionals and organizations in implementing blockchain transformation projects by providing an overview of current research, knowledge gaps, and future research directions. |

**RQ6:** How are large healthcare files currently secured in blockchain–cloud integration, and what methods and techniques are currently being utilized?

The intersection of healthcare data management, cloud computing, and blockchain technology represents a crucial nexus in the evolution of modern healthcare systems. As the healthcare industry embraces digital transformation, the secure storage and transmission of large healthcare files have become paramount. Blockchain–cloud integration emerges as a promising solution, combining the scalability and flexibility of cloud computing with the immutability and decentralized nature of blockchain [43].

These are some of the methods and technologies used nowadays to enhance this secure integration. Table 8 shows the overview of how large healthcare files are currently protected in the blockchain-cloud integration.

It is important to note that the specific methods and techniques employed may vary depending on the blockchain–cloud integration solution and the regulatory requirements of the healthcare sector. Additionally, a combination of these approaches is often used to provide a robust, multi-layered security framework for protecting large healthcare files.

Table 8. Landscape of how large healthcare files are currently secured in blockchain–cloud integration.

| Approaches | Works | Description |
|---|---|---|
| Encryption | [52,83] | Protecting patient privacy is paramount in healthcare. To ensure this, sensitive medical information is scrambled with powerful codes before being stored electronically. This scrambling process, known as encryption, acts like a complex lock. Even if someone manages to break into the storage system, the scrambled data remains unreadable, safeguarding patient information from unauthorized access. |
| Access control | [20,40,42,45,69,85,96,97] | Access to healthcare files is typically governed by robust access control mechanisms. Role-based access control (RBAC) and attribute-based access control (ABAC) are commonly used to regulate access based on predefined roles, permissions, and attributes. These mechanisms ensure that only authorized individuals or entities can access and manipulate the data. |
| Off chain | [44,103,104] | In many blockchain–cloud integrations, the actual healthcare files are stored off-chain (e.g., in cloud storage), while the metadata and hashes of the files are recorded on the blockchain. This approach leverages the immutability and transparency of the blockchain for data integrity and provenance tracking, while cloud storage provides scalability and efficient data retrieval. |
| Cryptographic techniques | [105–109] | SMPC is a cryptographic technique that enables multiple parties to jointly compute a function over their inputs while keeping the inputs private. In the context of healthcare data, SMPC can be used to perform computations or analysis on encrypted data without revealing the original data to any party involved. ZKPs are cryptographic protocols that allow one party (the prover) to prove to another party (the verifier) that a statement is true without revealing any additional information. In healthcare data management, ZKPs can be used to verify the integrity and correctness of data without exposing the actual data. |
| Oracles | [35,66,67,71] | Oracles are third-party services that serve as bridges between blockchains and external data sources or systems. They provide a secure and trusted way to bring off-chain data onto a blockchain network, enabling smart contracts and decentralized applications (DApps) to interact with real-world data and events. In essence, oracles act as data providers, feeding external information to the blockchain in a tamper-proof and verifiable manner. This external data can include various types of information, such as financial data, weather data, IoT sensor readings, or in the case of healthcare, medical records, test results, or patient information stored in cloud-based systems. |
| Secure Data Sharing and Consent Management | [23,73,80,87,96,110] | Blockchain–cloud integrations often implement secure data-sharing mechanisms that allow patients to control and grant access to their healthcare data. Consent management protocols ensure that data are shared only with authorized entities and for approved purposes, respecting patient privacy and data sovereignty. |

The choice of security measures and their implementation details will be influenced by factors such as the type of healthcare data being handled (e.g., electronic medical records, genomic data, medical imaging), the cloud service provider's infrastructure and security protocols, and the blockchain platform being integrated. Compliance with relevant healthcare regulations, such as HIPAA (Health Insurance Portability and Accountability Act) in the United States or GDPR (General Data Protection Regulation) in the European Union, is also a critical consideration.

Furthermore, as the landscape of blockchain–cloud integration for healthcare data management continues to evolve, new security challenges may arise, necessitating the

development and adoption of advanced techniques. For example, emerging technologies like homomorphic encryption, which allows computations to be performed on encrypted data without decrypting it, or secure multi-party computation (SMPC) protocols could play a more significant role in ensuring data privacy and enabling privacy-preserving analytics.

### 3.3. Innovative Research Perspectives

In this section, the paper delves into the vanguard of research at the nexus of those papers renowned for their groundbreaking contributions within the realm of blockchain technology, cloud computing, and healthcare. The selection process involved identifying six distinct categories deemed pivotal in advancing the discourse. Within each category, two seminal papers have been chosen for their exemplary innovation and impact. These categories include Encryption, Access Control, Off-Chain Cryptographic Techniques, Oracles, Secure Data Sharing, and Consent Management.

### 3.3.1. Encryption

**PriFR: Privacy-Preserving Large-Scale File Retrieval System via Blockchain for Encrypted Cloud Data** [52].

PriFR stands out as a comprehensive framework intertwining cloud computing and blockchain technology to create a robust and privacy-preserving file retrieval system. Its key strengths lie in facilitating numerical range queries using Order-Preserving Encryption (OPE) while incorporating differential privacy (DP) measures to bolster query accuracy. Additionally, PriFR implements encrypted keyword search functionality through Searchable Symmetric Encryption (SSE) methods, enhancing its versatility for secure data retrieval. The framework prioritizes privacy preservation by ensuring the confidentiality of both original file data and metadata, particularly notable in DP-enabled range query protocols. Through rigorous performance evaluations, PriFR demonstrates its efficiency and privacy trade-offs, shedding light on its computational complexity and resilience under various attack scenarios. Looking ahead, PriFR paves the way for future explorations into blockchain-enabled big data processing, setting a solid foundation for potential collaborations and extensions in this evolving domain.

**Keyword Searchable Encryption Scheme Based on Blockchain in Cloud Environment** [83].

The authors present a comprehensive approach to address the challenges of keyword search on encrypted data in cloud environments. It introduces a searchable encryption scheme that leverages Public Key Searchable Encryption (PEKS) and various search schemes to enable efficient and secure data retrieval without decrypting the data. By integrating blockchain technology, the scheme enhances security and traceability, ensuring the non-repudiation of user queries. The system model involves key entities such as Data Owner, Data User, Cloud Service, Trusted Third Party, and Blockchain. The proposed scheme guarantees semantic security against keyword attacks and offers traceability through blockchain, highlighting a novel and effective solution for secure data retrieval in cloud environments.

### 3.3.2. Access Control

**Ancile: Privacy-preserving framework for access control and interoperability of electronic health records using blockchain technology** [40].

The research of a blockchain-based framework is explored by giving the readers a perspective of how blockchain technology can be used to manage access control and the interoperability of electronic health records. The main contribution is that instead of storing entire records on the blockchain, Ancile only stores hashes of data references and sends the actual query links privately over HTTPS. It also employs advanced cryptographic techniques like proxy re-encryption to facilitate the secure transfer of EHRs and storage of keys/small records directly on the blockchain. This enhances privacy and scalability compared to previous blockchain EHR systems.

On the other hand, Ancile's focus on patient ownership and control over their EHR data is also being explored. It does not use mining incentives, operating on the premise that the data belongs to patients, and is not a currency to be exchanged. Access control is handled via smart contracts to account for the varying roles of patients, providers, and third parties. The permissioned blockchain uses a consensus algorithm rather than proof-of-work, allowing for better validation when managing nodes. These elements uphold patient ownership rights while enabling secure, role-based access management.

The next figure (Figure 3) explains how the authors planned to use Ancile to accomplish a secure interaction between the actors involved in medical issues and the blockchain.

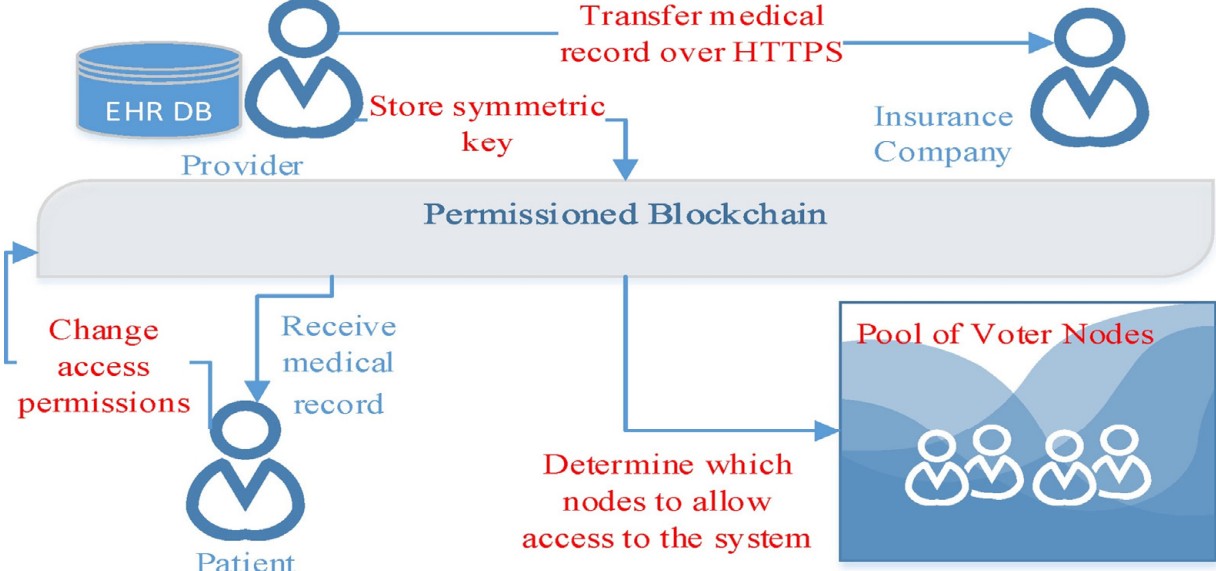

**Figure 3.** Ancile's ability for multiple parties to securely interact with the blockchain and its information [40].

The authors intend to convey that Figure 3 provides an overview of the potential interactions among various stakeholders using Ancile. Representing three key parties—patients, providers, and insurance companies—the diagram succinctly illustrates their connections with each other and blockchain technology. Notably, the red text highlights Ancile's distinctive features compared to common blockchain-based health data management systems, while the blue text denotes processes already proposed or existing in electronic health record (EHR) management.

**Blockchain-Based Access Control Model to Preserve Privacy for Personal Health Record Systems** [45].

The presented blockchain-based model (Figure 4) aims to overcome limitations like storage constraints, privacy issues, and performance bottlenecks when applying blockchain to personal health record systems. It achieves this by employing proxy re-encryption and other cryptographic techniques to preserve data privacy while utilizing a private blockchain for the tamper-resistant storage of metadata. The actual encrypted PHR data resides in cloud storage. The model provides fine-grained access control, consent revocability, auditability via the blockchain, and tamper resistance of records.

A remarkable aspect is the use of proxy re-encryption keys managed by the PHR owner (patient) to selectively share encrypted PHR data while retaining ownership control. This is facilitated by a semi-trusted gateway server that verifies authenticity, re-encrypts data for authorized users, and manages the private blockchain metadata. Detailed protocols outline the workflows for system setup, storing new PHRs, retrieving PHRs, and revoking user access with auditability on the blockchain.

The authors aim to convey that firstly, the paper introduces the system setup phase, elucidating how the initial framework of the system is established. Subsequently, it delin-

eates the step-by-step procedure for accessing personal health record (PHR) data within the system. Finally, it delves into the process of revoking access rights to PHR data.

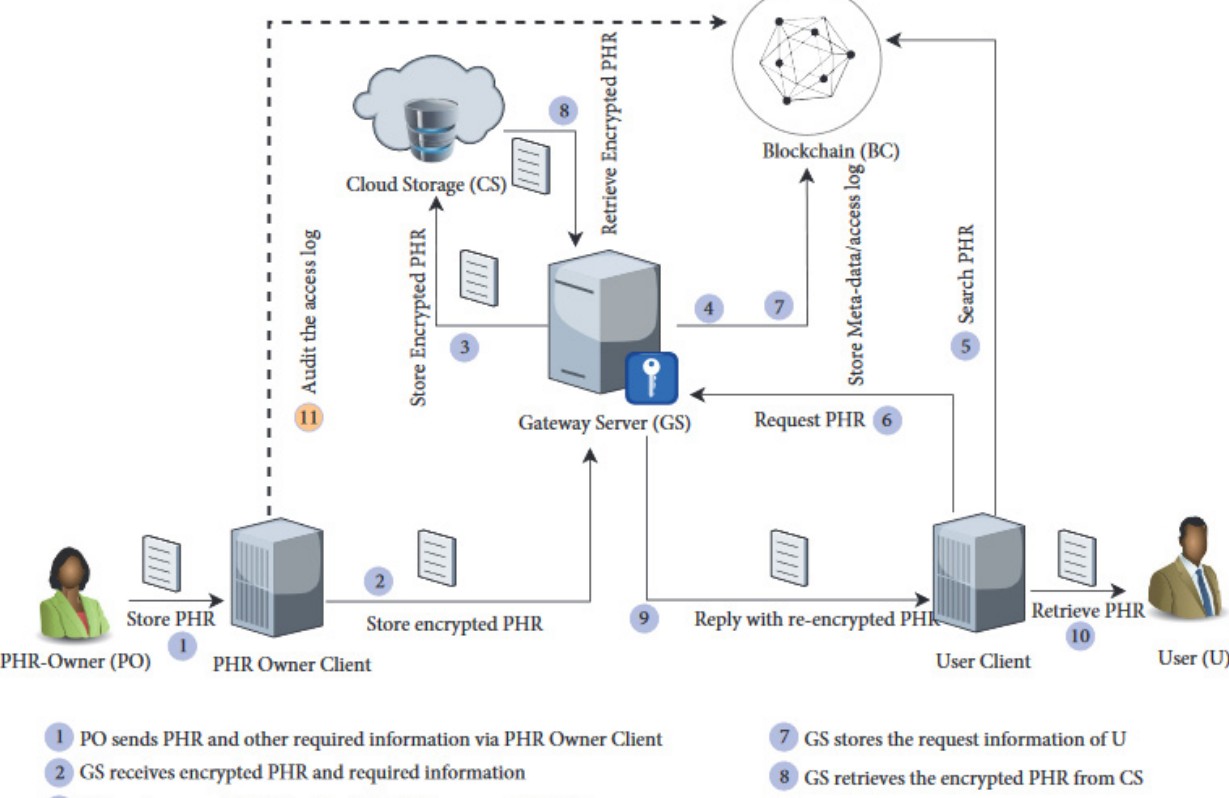

**Figure 4.** The system architecture [45].

### 3.3.3. Off Chain

**hOCBS: A privacy-preserving blockchain framework for healthcare data leveraging an on-chain and off-chain system design** [103].

The proposed hOCBS (Hybrid Off-Chain Blockchain System) framework is designed to leverage the advantages of both on-chain and off-chain components for managing healthcare data on blockchains. It presents three distinct reference models mapped to different types of healthcare data—protected health information, consumer health information, and genomic data (Figure 5). Each reference model accounts for the specific privacy and regulatory requirements governing that particular data type. The overall framework prioritizes a patient-centric approach that enables data ownership and stewardship by individuals over their own healthcare data.

The Genomic Data Reference Model (Figure 5) boasts an unparalleled level of modularity and adaptability, tailored to accommodate diverse user preferences. Off-chain storage options include distributed storage, personal storage, or the utilization of third-party services. To enhance security, a secondary layer of authentication is introduced through a phylogenetic off-chain storage mapping mechanism. This feature empowers researchers to solicit data access from owners while safeguarding sensitive information. Furthermore, a token-based system, coupled with a consumer-centric security framework, fosters a culture of genomic data sharing.

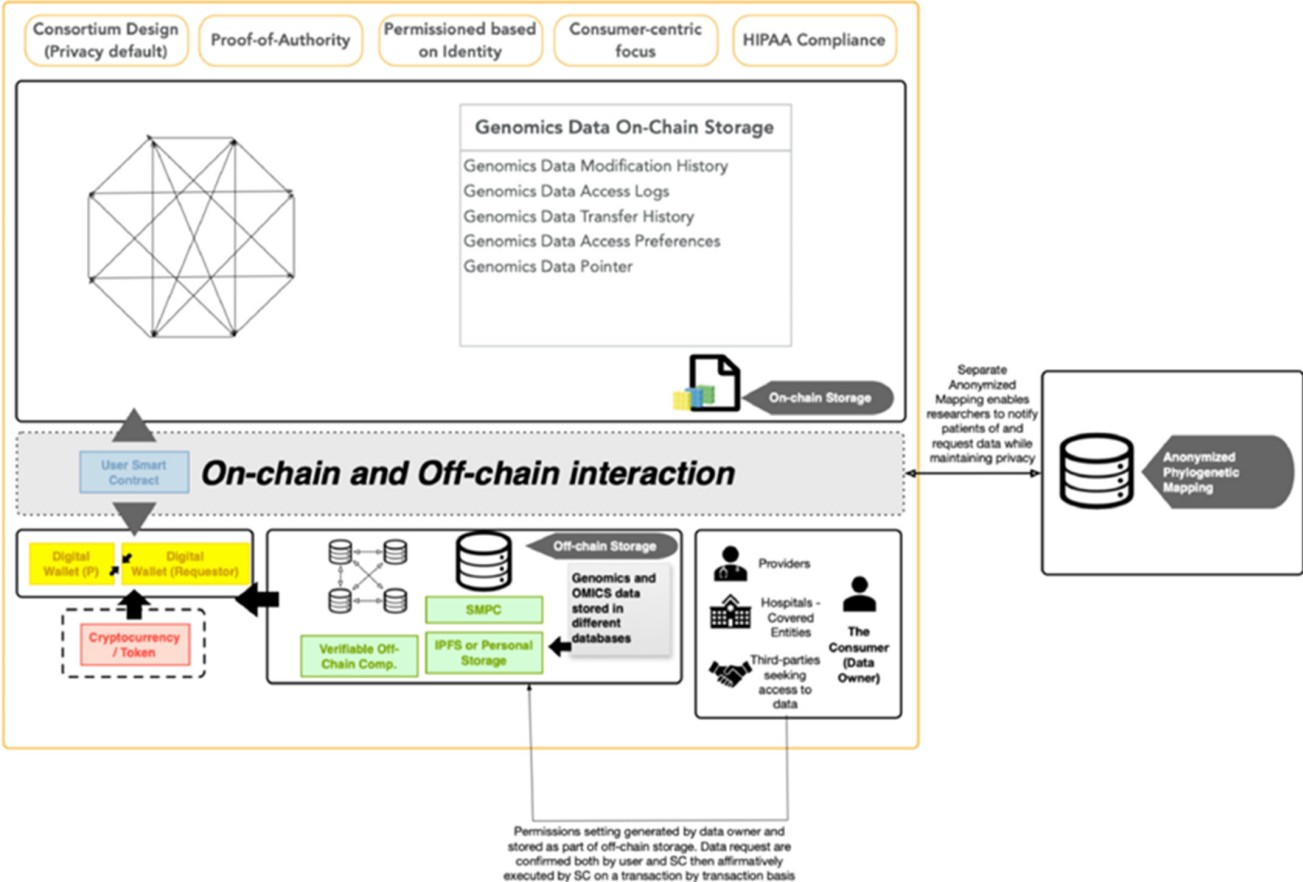

**Figure 5.** Anonymized and secure multi-factor consent processes [103].

Core blockchain features like smart contracts for access control, digital wallets for identity management, and tokens for incentivization are incorporated into the framework. The hybrid on-chain/off-chain architecture allows for the secure sharing of healthcare data, data sovereignty for patients, and enhanced trust while improving portability and transparency. On-chain components store metadata and enforce access policies, while off-chain storage holds the actual data. The key goal is to establish a privacy-preserving solution that can "liberate" healthcare data by allowing controlled sharing and citizen-driven data governance while ensuring compliance with relevant data protection regulations like HIPAA and GDPR.

**Scalable blockchain model using off-chain IPFS storage for healthcare data security and privacy** [44].

The proposed model presents a scalable and secure blockchain framework for managing electronic health records in healthcare. It integrates blockchain technology with the Interplanetary File System for the decentralized and off-chain storage of encrypted EHR data. The blockchain stores only the encrypted hashes and metadata, while the actual EHR files are stored on the distributed IPFS network (Figure 6). This hybrid approach aims to address blockchain's scalability limitations for large data storage. The framework enables a patient-centric access model where patients act as digital stewards, controlling access to their EHR data for healthcare providers on an as-needed basis.

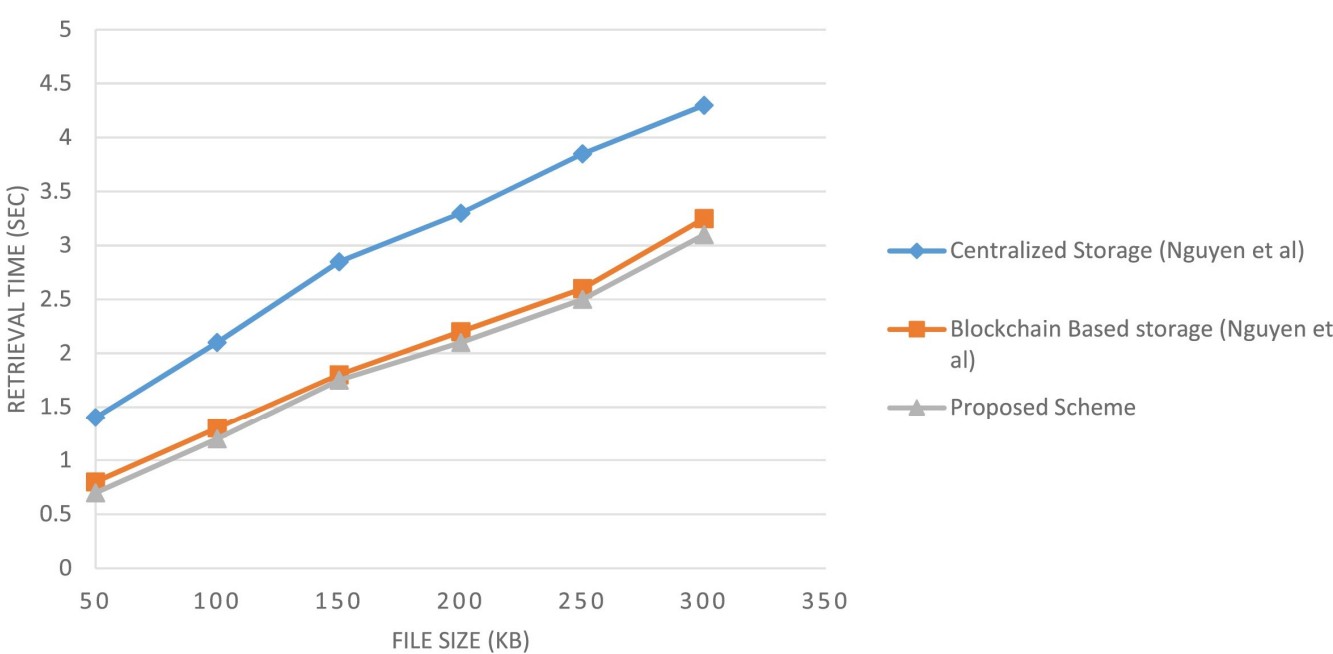

**Figure 6.** Comparison of storage retrieval time structure [44].

Multiple layers of security are implemented, including two/multi-factor authentication to prevent fake node attacks, symmetric encryption (AES-128) of EHR data before storing on IPFS, asymmetric encryption (RSA-4096) for sharing keys with authorized entities, digital signatures (RSA-1024) for transaction validation, and SHA-256 hashing of encrypted data. The off-chain IPFS storage protects data privacy by ensuring that even if encrypted files are retrieved, they cannot be decrypted without authorized access. The proof-of-work consensus model aims to prevent Sybil attacks on the blockchain network. Overall, the integrated blockchain-IPFS model targets secure EHR sharing while preserving patient data ownership and privacy.

### 3.3.4. Cryptographic Techniques

**Using Secure Multi-Party Computation to Protect Privacy on a Permissioned Blockchain** [105].

The authors present a comprehensive framework for a publicly verifiable, secure multi-party computation (MPC) protocol tailored for Hyperledger Fabric environments. The protocol is designed to ensure data privacy, correctness, and verifiability in smart contract execution. Through a combination of homomorphic encryption and secret sharing, participants encrypt their inputs using different public keys, ensuring that each participant only sees their authorized share, thus safeguarding input privacy. The protocol also incorporates mechanisms to resist collusion attacks, with a threshold that remains secure even with a conspiracy among up to n-1 participants. Public verifiability is achieved through commitments, allowing anyone to check the correctness of the computations.

Experimental results demonstrate the impact of key size on running time, with larger key sizes providing increased security at the cost of longer response times. Comparison with other schemes reveals the competitive performance of the proposed solution, particularly in smaller networks. However, scalability is noted as a potential limitation due to communication costs, suggesting avenues for future work in optimizing communication efficiency and exploring more efficient P2P communication channels. The overall conclusion highlights the protocol's effectiveness in ensuring privacy, correctness, and verifiability in smart contract execution within permissioned blockchain environments,

laying the groundwork for practical applications such as the statistical analysis of sensitive data with guaranteed privacy protections.

**CRYPTEN: Secure Multi-Party Computation Meets Machine Learning** [106].

The paper provides a comprehensive overview of CRYPTEN, a framework designed for secure multi-party computation (MPC) in machine learning. One of the key findings is the significant performance difference between CRYPTEN and PyTorch, particularly in terms of inference time and communication overhead. Benchmarks on various tasks, including text classification, speech recognition using the Wav2Letter model, and image classification using ResNet-18 and ViT-B/16 models, reveal that CRYPTEN is several orders of magnitude slower than PyTorch, mainly due to the encryption and communication protocols inherent in secure MPC. Despite this performance gap, CRYPTEN offers valuable capabilities for privacy-preserving machine learning, such as secure arithmetic and binary secret sharing, private addition and multiplication operations, and support for linear and non-linear functions crucial in machine learning models.

Moreover, the paper highlights the challenges and trade-offs associated with secure MPC, such as numerical issues in fixed-point representations and the need for differential privacy mechanisms to mitigate information leakage.

**Leveraging zero-knowledge proofs for blockchain-based identity sharing: A survey of advancements, challenges, and opportunities** [107].

The paper provides a detailed comparison between ZK-SNARKs and ZK-STARKs, highlighting key differences and advantages in various aspects. ZK-SNARKs and ZK-STARKs both excel in privacy preservation, allowing identity attributes to be proven without disclosing sensitive data, a crucial feature for privacy-sensitive sectors like healthcare and finance. However, ZK-STARKs showcase superior efficiency with smaller proofs and eliminate the need for a trusted setup, enhancing scalability and cost-effectiveness in blockchain applications. In terms of security, ZK-STARKs stand out by removing the reliance on a trusted setup, which is especially relevant as blockchain technology evolves and quantum computing threats emerge. Furthermore, ZK-STARKs exhibit transparency in their construction, making them ideal for public blockchains, and ensuring trust in the verification process. While ZK-SNARKs enjoy wider adoption and standardization, ZK-STARKs are gaining attention for their security and efficiency benefits, suggesting a potential shift toward their adoption, particularly in applications demanding higher security and efficiency. These findings underscore the evolving landscape of zero-knowledge proofs and their significant role in shaping secure and decentralized identity sharing on blockchain platforms.

**Neural Fairness Blockchain Protocol Using an Elliptic Curves Lottery** [109].

The paper presents a comprehensive exploration of blockchain consensus algorithms, focusing on the novel Neural Fairness Protocol (NFP) and its comparison with established methods. The primary goal of consensus algorithms in blockchain is to ensure the steady functioning of the technology, where nodes agree on specific values or transactions. The study emphasizes the importance of fairness in achieving a balanced consensus, particularly in avoiding situations where a few nodes hold disproportionate consensus power. The NFP, designed to address these challenges, offers high throughput and fairness by utilizing Conflict of Interest (CoI) detection through a Conflict Graph (CG). By leveraging utility factors associated with nodes and employing the Maximum Weight Independent Set (MWIS) algorithm, the NFP facilitates a fair selection process for consensus committee members. Furthermore, the paper highlights the role of rewards and incentives in maintaining a vibrant blockchain ecosystem, aligning with Satoshi Nakamoto's view that nodes contributing to the network should be incentivized. Through reward mechanisms tied to block production and fairness activities, the NFP ensures that committee members receive their fair share of tokens, promoting continued participation and network integrity. The study concludes by emphasizing ongoing research and development efforts to enhance the confidentiality, security, and efficacy of the NFP, positioning it as a promising advancement in blockchain consensus protocols.

**Blockchain and Demand Response: Zero-Knowledge Proofs for Energy Transactions Privacy** [108].

The paper explores a comprehensive framework for ensuring privacy and security in decentralized Demand Response (DR) programs using blockchain technology. It introduces a novel approach where prosumer energy data privacy is maintained through the integration of zero-knowledge proofs (ZKPs) within smart contracts. This integration allows for the validation of energy consumption without exposing sensitive data to the public blockchain. Key findings indicate that this approach not only safeguards prosumer privacy but also enables the efficient validation and detection of deviations from the requested flexibility, which is crucial for maintaining program integrity. Additionally, the paper addresses challenges such as the risk of private key theft and scalability concerns due to increased gas consumption from ZKPs. It suggests solutions like specialized hardware for secure key management and proposes optimizations in transaction registration intervals to manage scalability efficiently. Overall, the research underscores the potential of blockchain and ZKP integration in enhancing privacy, security, and transparency in DR programs, paving the way for more robust and reliable energy management systems.

### 3.3.5. Oracles

**A reinforcement learning model for the reliability of blockchain oracles** [35].

The paper proposes BLOR, a Bayesian Bandit Learning model for Oracles Reliability, to identify trustless and cost-efficient oracles for smart contracts on blockchain networks. BLOR integrates a novel Bayesian cost-dependent reputation model with reinforcement learning techniques like the Knowledge Gradient algorithm. The BCRM learns the reliability of oracles over time based on their performance history, while considering their cost. The KG algorithm guides the selection of oracles that balance short-term rewards with long-term knowledge gain about their reliability.

BLOR is implemented on the Ethereum blockchain, with most computations executed on-chain to enable verification by validators. It utilizes a decentralized random number generator involving all validators to address the issue of achieving consensus on probabilistic computations. The selected oracles perform the requested task, and BLOR updates their reputation values, pays them accordingly, and informs the requester, ensuring a trustless and cost-optimized oracle management system. Benchmarking shows BLOR's steady performance in identifying reliable and economical oracles, even in highly noisy environments. Figure 7 delineates the workflow of BLOR, designed to select the most reliable and cost-effective oracles.

The majority of this process occurs on chain, facilitating validation and consensus among the blockchain's validators to mitigate bias. Considering the insurance company as the service provider and the customer as the service requester, transactions are executed via a smart contract. Upon initiation of a claim request by the service requester through the smart contract, BLOR orchestrates the selection of oracles tasked with processing the request. Initially, the smart contract generates a new contract involving a network of oracles, serving as the secondary beneficiary party. This contract encompasses a fresh set of rules and conditions, including payment and compensation policies.

**From trust to truth: Advancements in mitigating the Blockchain Oracle problem** [67].

The paper presents an analysis of the blockchain oracle problem, focusing on the critical role oracles play in bridging smart contracts with off-chain, real-world data for decentralized applications. A major contribution is providing a thorough taxonomy and classification of different types of blockchain oracles based on their design patterns, trust models, underlying architectures, and consensus mechanisms. It summarizes the tasks and responsibilities of oracles across various blockchain-based systems and domains, including healthcare applications. The paper examines how trust is established and maintained in these different oracle solutions through techniques like voting, reputation systems, and incentive mechanisms.

Furthermore, the paper highlights the issues, trade-offs, and challenges associated with using blockchain oracles, such as centralization risks, single points of failure, and the compromise of a trustless operation. It reviews and analyzes some of the significant existing oracle implementations like Chain-link and Provable. The authors carry out a gap analysis, identifying areas that lack sufficient research, and outline future directions to advance blockchain oracles toward achieving robust, secure, and truly decentralized operations. A key focus is on oracle solutions tailored for healthcare and medical data, given the sensitivity and critical importance of this domain.

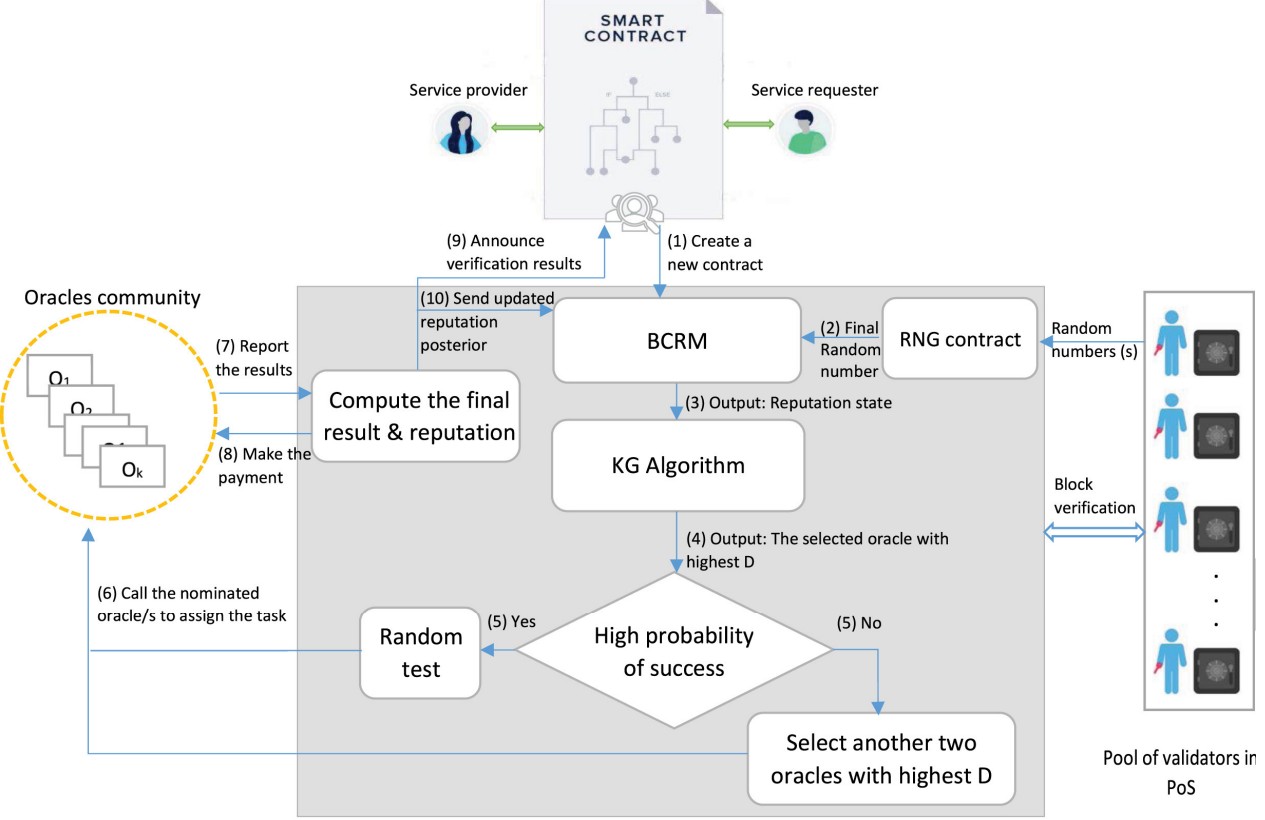

**Figure 7.** The system architecture [35].

3.3.6. Secure Data Sharing and Consent Management

**BBDS: Blockchain-based data sharing for electronic medical records in cloud environments** [23].

The paper outlines a blockchain-based data sharing scheme for electronic medical records systems, emphasizing secure data sharing and privacy protection. The block structure, crucial to ensuring immutability, includes a format with a unique identifier, block size, and header hashed with sha256 (sha256()). This header contains essential elements like version number, previous block hash, Merkle root hash, timestamp, target difficulty, and nonce. User requests trigger block creation, with transactions including timestamps, user IDs, and signatures, ensuring validation and integrity. Protocols between users, issuers, and verifiers facilitate authentication, membership verification, and key generation, ensuring secure access and data handling. The proposed system addresses scalability concerns, with simulations indicating efficient data processing and manageable blockchain growth. In conclusion, the paper presents a robust framework for secure medical data sharing, highlighting the importance of blockchain technology in maintaining data integrity and privacy.

**A blockchain-based attribute-based signcryption scheme to secure data sharing in the cloud** [73].

The paper introduces a novel approach, Blockchain-based Attribute-Based Signcryption (BABSC), aimed at enhancing the security of data sharing in cloud environments. By combining blockchain technology with attribute-based signcryption, the proposed scheme addresses several key challenges faced by traditional cloud data-sharing systems. These challenges include reliance on trusted third parties, centralized data storage, high operational costs, and security concerns. BABSC offers efficient access control over data stored in cloud servers, ensuring confidentiality and unforgeability. The use of attribute-based signcryption enables secure access control, data confidentiality, and ciphertext unforgeability, while blockchain technology provides a decentralized and immutable ledger for recording transactions. The paper further compares BABSC with existing ABSC schemes, demonstrating its superior performance in terms of storage and computations. Overall, BABSC presents a promising solution to secure data sharing in cloud environments, mitigating the risks associated with centralized storage and enhancing user control and privacy.

**Novel Blockchain and Bi-Linear Polynomial-Based QCP-ABE Framework for Privacy and Security over the Complex Cloud Data** [110].

The research paper presents a novel approach for securing sensitive data in cloud environments through a combination of advanced cryptographic techniques. One key contribution is the integration of bi-linear maps and chaotic key generation processes, which enhance the privacy and security of cloud-stored information. By utilizing a group of bi-linear polynomial curves and randomized complex chaotic utilities, the proposed model effectively addresses the limitations of traditional Attribute-Based Encryption (ABE) approaches. This new model demonstrates improved accuracy and efficiency in key generation, encryption, and decryption processes, requiring less computational overhead compared to existing methods like CPABE, CQ-CPABE, KPABE, and QCP-ABE. The experimental results validate the efficacy of the proposed model, showcasing its ability to handle large volumes of sensitive data while maintaining high levels of security. Moreover, the paper outlines potential future directions, suggesting advancements in encryption and decryption processes using deep learning frameworks for multi-document file formats without compromising data quality or resolution. Overall, the research contributes significantly to the ongoing efforts to secure cloud-based data storage and communication, with implications for improving privacy and confidentiality in digital ecosystems.

## 4. Discussion

This comprehensive review of almost 100 recent papers provides valuable insights into the current landscape and progress of secure blockchain–cloud integration for healthcare data management. Our analysis reveals several notable themes and trajectories that warrant further discussion.

The systematic review establishes a foundation for future research directions, as highlighted in [101]. The identified research gaps, coupled with the proposed future topics, can guide scholars and academics interested in advancing the field of blockchain security applications. This foresight adds depth to the ongoing discourse on the evolution of blockchain technology.

In summary, the integration of blockchain and cloud computing is a complex field presenting challenges and opportunities. There is a clear need for a balanced approach to address economic, technical, and security aspects. The dialogue among different authors provides a comprehensive insight that can guide future research toward more robust and effective solutions.

### 4.1. Growing Recognition and Core Themes

Overall, the steady increase in publications over recent years highlights the growing recognition of the potential benefits between blockchain and cloud technologies. The predominant focus on security, integration approaches, and chain integrity underscores the importance of these factors in realizing the benefits of this merger.

The findings corroborate conclusions from previous seminal works regarding the role of blockchain in addressing vulnerabilities introduced by cloud computing through enhanced transparency, traceability, decentralization, and automation [29]. The reviewed papers also align with earlier studies emphasizing the need to balance blockchain's security with the performance and efficiency gains enabled by the cloud [25].

### 4.2. Challenges and Innovation

However, substantial challenges remain when it comes to practical implementation. Scalability is a persistent issue, as the volume of healthcare data continues to expand exponentially [24]. While some proposed solutions aim to improve transaction efficiency, network congestion and lag times persist. The lack of common standards and protocols also hinders interoperability between diverse blockchain implementations and cloud providers [2].

This review reveals a range of innovative techniques and architectures seeking to overcome these hurdles, from decentralized oracles [35] to bioacoustics authentication mechanisms [26]. However, regulatory uncertainties around emerging blockchain models present additional complications [8]. More research is needed to determine optimal frameworks that comply with data protection regulations.

### 4.2.1. Secure Protocols Used for Secure Connections via Cloud and Blockchain

At present, there is a variety of protocols and rules that are being used to achieve a secure connection between blockchain and cloud; for this reason, we describe the current landscape of secure protocols utilized for establishing connections via cloud computing and blockchain technology in the next few lines. As the demand for secure data transmission and storage continues to surge in today's digital era, understanding the protocols at the forefront of ensuring confidentiality, integrity, and authenticity becomes paramount. From encryption standards to access control mechanisms, this section navigates through the intricate web of protocols that underpin secure communication in cloud and blockchain environments. By dissecting the intricacies of these protocols, the authors aim to provide insights into their functionality, strengths, and limitations, ultimately shedding light on the evolving paradigms of secure connectivity in the modern digital ecosystem.

### Cloud Access Nodes

Cloud access nodes are dedicated gateways deployed in the cloud that act as secure bridges between blockchain networks and cloud services. They facilitate encrypted data transfer, perform validations, enforce access controls, and translate protocols, enabling seamless and secure integration between the two environments [20,85].

- These are typically virtual machines or containers deployed in the cloud environment, acting as dedicated gateways or proxies.
- They often run specialized software or agents that handle secure communication and data transfer between the blockchain network and cloud services.
- The cloud access nodes can be part of a load-balanced cluster or pool for high availability and scalability.

  **They can implement various security controls, such as:**

- Data encryption and decryption using secure algorithms like AES, RSA, or ECC (Elliptic Curve Cryptography) [3,111];
- Digital signature verification using algorithms like ECDSA or RSA to ensure data integrity and authenticity [58];
- Access control and authorization mechanisms, such as OAuth 2.0, JWT, or custom authentication protocols [96];
- Firewalls, intrusion detection/prevention systems, and other security appliances for protecting the nodes themselves [74].

The nodes can also handle protocol translation, data format conversion, and other integration tasks between the blockchain and cloud environments [109].

They can be configured to support different blockchain protocols (e.g., Ethereum, Hyperledger Fabric) and cloud service APIs (e.g., AWS, Azure, GCP).

Secure APIs

Secure APIs provide a standardized and encrypted communication interface for blockchain networks and cloud services to interact. They use protocols like HTTPS/TLS, along with authentication and authorization mechanisms like OAuth 2.0 or JWT, ensuring secure data exchange and service invocation between the two environments [112].

- APIs are the communication interfaces that enable different systems or applications to interact with each other.
- In the context of blockchain and cloud integration, secure APIs are used to facilitate the secure data exchange and invocation of services between the two environments.

**Common secure API protocols include the following:**

- REST APIs over HTTPS/TLS: REST (Representational State Transfer) is a widely used architectural style for building web services. HTTPS/TLS provides encryption and authentication for data in transit [113].
- gRPC over TLS: gRPC is a modern, open-source remote procedure call (RPC) framework that uses HTTP/2 for transport and can be secured with TLS [114].
- WebSocket over TLS: WebSocket enables real-time, bidirectional communication between client and server, and can be secured with TLS [115].

**Authentication and authorization mechanisms used with secure APIs include the following:**

- OAuth 2.0: An industry-standard protocol for authorization, allowing third-party applications to obtain limited access to resources on behalf of a user [116].
- JSON Web Tokens (JWT): A compact, URL-safe means of representing claims (e.g., user identity, permissions) securely between parties [117].
- API keys and secrets: Unique identifiers and secrets issued by the API provider for authentication and authorization purposes [118].
- Certificate-based authentication: Using X.509 digital certificates for mutual authentication between client and server [119].

Data Encryption

Data encryption is the process of converting plaintext data into ciphertext using secure encryption algorithms and keys before transferring it between blockchain and cloud. Common algorithms like AES, RSA, and ECC are used to protect data confidentiality and integrity during transit [111].

- Data encryption is the process of converting plaintext data into ciphertext using an encryption algorithm and a key.

**Common encryption algorithms used for data transfer between blockchain and cloud include the following:**

- AES (Advanced Encryption Standard): A symmetric-key algorithm widely used for encrypting data in transit and at rest. AES-256 is a common variant with a 256-bit key size [111].
- RSA (Rivest–Shamir–Adleman): A popular public-key cryptography algorithm used for secure data transmission and digital signatures [111].
- ECC (Elliptic Curve Cryptography): A type of public-key cryptography based on the algebraic structure of elliptic curves, offering strong security with smaller key sizes compared to RSA [111].

**The encryption process typically involves the following:**

- Generating a secure encryption key (symmetric or asymmetric) using industry-standard key generation algorithms;
- Encrypting plaintext data using the encryption algorithm and key;
- Securely transmitting the encrypted data (ciphertext) to the destination (blockchain or cloud).

At the receiving end, the ciphertext is decrypted using the appropriate decryption key to obtain the original plaintext data.

Digital Signatures

Digital signatures, based on public-key cryptography, are used to verify the integrity and authenticity of data transferred between blockchain and cloud. Algorithms like ECDSA and RSA are employed to generate and validate digital signatures, ensuring data have not been tampered with and originated from the claimed source [120].

- Digital signatures are used to provide data integrity, authentication, and non-repudiation for data transfers between blockchain and cloud environments.
- They are based on public-key cryptography, where each party has a public/private key pair.

**The process typically involves the following:**

- Calculating a cryptographic hash (e.g., SHA-256, SHA-3) of the data to be signed;
- Using the private key and a digital signature algorithm (e.g., ECDSA, RSA-PSS) to generate a digital signature from the hash;
- Transmitting the data and the digital signature to the recipient;
- The recipient calculates the hash of the received data and verifies the digital signature using the sender's public key and the same signature algorithm.

If the signature is valid, the recipient can be assured that the data have not been tampered with (integrity) and originated from the claimed sender (authentication and non-repudiation).

Secure Channels

Secure channels, such as TLS/SSL, IPsec, or VPNs, provide end-to-end encryption and authentication for the entire communication session between the blockchain and cloud environments. They ensure data confidentiality, integrity, and protection against eavesdropping and man-in-the-middle attacks [121–123].

- Secure channels encrypt the entire communication session or connection between the blockchain and cloud environments, providing end-to-end protection for data in transit.

**Common protocols used for secure channels include the following:**

- TLS/SSL (Transport Layer Security/Secure Sockets Layer): These protocols establish an encrypted and authenticated secure channel over an insecure network (e.g., the internet). TLS 1.2 and 1.3 are the latest versions [123].
- IPsec (Internet Protocol Security): A suite of protocols that provide secure communication at the network layer (Layer 3) of the OSI model, enabling secure communication between networks or hosts [121].
- VPNs (Virtual Private Networks): VPNs create a secure, encrypted tunnel over an untrusted network, effectively extending a private network across a public network [122].

**Secure channel establishment typically involves the following:**

- Negotiating cryptographic parameters (e.g., cipher suites, key exchange algorithms) between the parties;
- Authenticating the parties using digital certificates or pre-shared keys;
- Performing a secure key exchange to derive a session key for encrypting the communication;
- Encrypting and authenticating all data transmitted over the secure channel using the negotiated parameters and session key.

Authentication and Authorization

Authentication mechanisms, like usernames/passwords, digital certificates, or biometrics, are used to verify the identities of users, devices, or systems accessing blockchain and cloud resources. Authorization techniques, such as RBAC, ABAC, and ACLs, control and restrict access to specific resources or operations based on predefined policies [124].

- Authentication is the process of verifying the identity of a user, device, or system trying to access a resource or service.
- Authorization is the process of determining what actions or resources an authenticated entity is allowed to access or perform.

**Authentication methods commonly used in blockchain and cloud integrations include the following:**

- Username and password credentials, often combined with multi-factor authentication (MFA) for added security [125];
- Digital certificates and public-key infrastructure (PKI) for machine-to-machine authentication [126];
- Biometric authentication methods like fingerprint, facial recognition, or iris scanning [127];
- Federated identity providers (e.g., OAuth, SAML) for single sign-on (SSO) and identity federation across systems [128].

**Authorization techniques include the following:**

- Role-based access control (RBAC), where permissions are assigned based on an entity's role within the organization [124];
- Attribute-based access control (ABAC), where access decisions are made based on attributes or characteristics of the requesting entity and the resource [124];
- Access Control Lists (ACLs) that explicitly define which entities have access to specific resources or operations [124];
- Policy-based access control, where access is governed by a set of rules or policies defined by the organization [124].

Monitoring and Auditing

Monitoring systems continuously track and log all data transfers, access attempts, and activities between blockchain and cloud, enabling timely detection and response to security incidents. Auditing maintains detailed logs for forensic analysis, compliance reporting, and regulatory requirements [129].

- Monitoring systems continuously track and log all data transfers, access attempts, and other activities between the blockchain and cloud environments.
- This helps detect and respond to security incidents, data breaches, or other anomalies in a timely manner.
- Automated alerting and incident response workflows triggered by predefined rules or anomaly detection models.

**Monitoring techniques include the following:**

- Network traffic monitoring and analysis using tools like network taps, packet captures, and intrusion detection/prevention systems (IDS/IPS).
- Application-level monitoring and logging of API calls, service invocations, and data transfers.
- Continuous monitoring of system logs, audit trails, and other security-related events.

*4.3. Promising Trajectories*

Looking ahead, several promising trajectories can be discerned from this review. Hybrid on-chain/off-chain architectures could help address scalability limitations [25]. Advances in cryptography, trusted execution environments, and zero-knowledge proofs

may expand privacy-preserving capabilities [30]. And decentralized storage networks with built-in incentives could reduce reliance on third-party cloud providers [95].

### 4.4. Integration Considerations from Reviewed Papers

The discussion surrounding the integration of blockchain and cloud computing encompasses critical aspects as presented in the reviewed papers. In Paper [33], emphasis is placed on the myriad architectures and models for this integration, recognizing the importance of adaptability to diverse contexts. However, Paper [49] raises concerns regarding potential interoperability challenges stemming from this diversity.

### 4.5. Architectures, Security, and Performance

There is a convergence of opinions in Papers [33,49] regarding security. Both acknowledge the significance of addressing security challenges in cloud service management and decentralized storage.

Concerning performance, Paper [33] underscores improvements that cloud computing can offer to blockchain, while the authors of Paper [49] pose challenges that the decentralized nature of IPFS faces, raising scalability concerns.

### 4.6. Applications and Economic Considerations in Healthcare

Papers [2,60,102] extend the focus toward the application of blockchain in the healthcare sector. The authors of Paper [97] highlight advancements and challenges in incorporating blockchain in healthcare, emphasizing its benefits in ensuring data integrity and patient privacy. However, Paper [98] delves into the intricacy of medical procedures and large-scale data management through blockchain, highlighting potential associated costs.

In terms of costs, the authors of Paper [98] conduct an economic feasibility analysis and conclude that their proposal is practical and efficient. However, delving deeper into the economic considerations in the integration of blockchain and cloud from the perspective of other authors would be insightful.

### 4.7. Challenges, Blockchain Types, and Consensus Mechanisms

All papers recognize challenges in integrating blockchain and cloud computing, such as security issues, interoperability, and scalability. Identifying areas for improvement and providing recommendations for future research are consistent across [33,49,60,97,98].

The categorization of blockchain into public, private, and consortium types, as discussed in [101], has implications for the design and deployment of blockchain solutions. Understanding the selection criteria for these types can provide valuable insights for practitioners and researchers, reinforcing the importance of tailoring blockchain architecture to specific use cases.

Consensus mechanisms, explored in [49,101], are pivotal in maintaining a fair and decentralized network. The comparison of different algorithms, such as proof-of-work (PoW) and Practical Byzantine Fault Tolerance (PBFT), sheds light on the trade-offs involved in selecting consensus mechanisms. This aligns with the scalability discussions in [49,98,101], emphasizing the need for efficient and sustainable blockchain solutions.

### 4.8. Technological Landscape for Secure Blockchain–Cloud Integration in Healthcare

The secure integration of cloud and blockchain technologies for healthcare data management relies on a foundation of complementary approaches that work together to enhance data security.

One key comparison is between encryption and access control mechanisms. Encryption safeguards data at rest, while access control mechanisms regulate who can access data in use. Another important comparison involves off-chain cloud storage for scalability and efficiency with large datasets, and cryptographic techniques like secure multi-party computation and zero-knowledge proofs for protecting sensitive data privacy [35,66,67,71].

Oracles act as bridges between the blockchain and external data sources like the cloud, enabling smart contracts to interact with real-world healthcare data. Secure data sharing with consent management mechanisms allows patients to control access to their health data [35,66,67,71].

Successful integration requires strategically combining the appropriate technologies based on specific use case requirements. Additional considerations include ensuring interoperability between different blockchain platforms and cloud providers, establishing clear regulatory frameworks, and continuous research and development.

## 5. Conclusions

This comprehensive review aimed to examine the current landscape and state of secure integration between blockchain technology and cloud computing for managing large healthcare files. Through a systematic analysis of almost 100 recent research papers, key insights have been synthesized to provide an overview of the progress, open challenges, and future trajectories related to this merger. The study reveals a steady increase in publications on blockchain–cloud integration over the past five years, with a heavy focus on security, integration approaches, and chain integrity. While blockchain shows promise in addressing vulnerabilities introduced by centralized cloud architectures, substantial challenges persist around efficiency, scalability, interoperability, and regulatory compliance.

The reviewed papers highlight innovative techniques and architectures that could help overcome these hurdles. Hybrid on-chain/off-chain designs, advances in cryptography and trusted execution environments, and decentralized storage networks represent promising pathways forward. More research is still needed to optimize solutions that balance scalability and efficiency with the security and privacy assurances of blockchain technology. Our analysis outlines key challenges and opportunities that can inform future research and development in this domain. Overall, the integration of these technologies shows considerable potential, but ongoing work is required to enable the practical realization of secure, decentralized, and performant systems for healthcare data management.

### 5.1. Key Findings

This section outlines the primary insights gleaned from the comprehensive review of the recent literature on blockchain–cloud integration for healthcare data management. The findings highlight the significant advancements, technological innovations, and the increasing scholarly interest in leveraging blockchain technology to enhance the security, scalability, and efficiency of cloud-based healthcare systems.

#### 5.1.1. Data Security and Privacy

Blockchain technology has emerged as a promising solution to enhance data security and privacy in healthcare. By leveraging its decentralized and immutable nature, blockchain ensures data integrity and transparency, reducing the risk of unauthorized access and tampering. Decentralized storage mechanisms further bolster data privacy by distributing patient data across a network, mitigating the vulnerabilities associated with centralized repositories. Additionally, the integration of blockchain and cloud computing facilitates improved data accessibility and interoperability among healthcare stakeholders, enabling seamless and secure data sharing.

- **Increased Data Security through Blockchain:** The implementation of blockchain in the healthcare data management system significantly increases data security by ensuring immutability and transparency. The study demonstrated that blockchain's cryptographic techniques effectively prevent unauthorized access and tampering, addressing critical security vulnerabilities inherent in traditional centralized systems.
- **Improved Data Accessibility and Interoperability:** The integration of blockchain and cloud computing facilitates better data accessibility and interoperability among different healthcare stakeholders. The study highlighted that seamless and secure

data sharing between providers, patients, and insurers is achievable through smart contracts and shared ledgers, improving coordination and patient outcomes.

- **Enhanced Data Privacy with Decentralized Storage:** Utilizing decentralized storage mechanisms provided by blockchain technology enhances data privacy. This research confirmed that patient data can be stored in a distributed manner, making it less vulnerable to single points of failure and large-scale breaches.

### 5.1.2. Academic Trends and Challenges

The increasing scholarly interest in blockchain and cloud integration within the healthcare sector reflects a growing recognition of its potential benefits. However, alongside this interest come significant challenges that need to be addressed. Scalability and performance issues arise due to the computational demands of blockchain transactions, while economic and resource implications pose obstacles to widespread adoption. Regulatory compliance considerations, such as aligning with frameworks like HIPAA and GDPR, add further complexity to implementation efforts.

- **Growing Academic Interest and Importance:** Analysis of recent research papers shows a substantial increase in scholarly attention toward integrating blockchain and cloud technologies for securing large healthcare files. Since 2017, the number of publications on this topic has steadily risen, peaking in 2022. This trend reflects the escalating recognition of the potential benefits and challenges associated with this integration.
- **Challenges in Scalability and Performance:** Despite the advantages, the research identified significant challenges in terms of scalability and performance. The paper detailed how integration can lead to increased latency and processing times due to the computational demands of blockchain transactions, which need to be addressed for practical, large-scale applications.

### 5.1.3. Technological Advancements and Patient-Centered Care

Despite challenges, ongoing technological advancements are driving innovation in blockchain–cloud integration for healthcare. Hybrid architectures, advancements in cryptography, and the emergence of decentralized storage networks offer promising solutions to overcome existing hurdles. These advancements pave the way for more personalized and patient-centered care approaches, empowering individuals with greater control over their health data and treatment plans.

- **Technological Advancements and Innovative Models:** The reviewed literature highlights promising technological advancements and innovative models aimed at overcoming existing hurdles. Hybrid on-chain/off-chain architectures, advances in cryptography and trusted execution environments, and the emergence of decentralized storage networks represent significant strides in addressing scalability, efficiency, and security concerns.
- **Potential for Patient-Centered Care:** The integration of these technologies has significant potential to enhance patient-centered care. The research showed that patients gain greater control over their health data, which can lead to more personalized and effective treatment plans.

### 5.1.4. Emerging Trends and Multidisciplinary Collaboration

Emerging trends in blockchain–cloud integration for healthcare include scalable consensus mechanisms, interoperability standards, and the development of quantum-resistant cryptographic solutions. However, realizing the full potential of these trends requires multidisciplinary collaboration. Blockchain experts, healthcare professionals, policymakers, and other stakeholders must work together to address the complex challenges and drive meaningful innovation in the field.

- **Emerging Trends and Research Directions:** The analysis also reveals emerging trends and promising research directions within the domain of blockchain–cloud integration for healthcare. Notable trends include the exploration of scalable consensus mechanisms tailored for enhanced scalability without compromising security, the development of interoperability standards to facilitate seamless data exchange, and the quest for quantum-resistant cryptographic solutions.
- **Multidisciplinary Collaboration:** The review underscores the need for multidisciplinary collaboration between blockchain experts, healthcare professionals, cloud service providers, policymakers, and other stakeholders to address the complex challenges and realize the full potential of this integration.

### 5.1.5. Implementation Challenges

While theoretical models and simulations demonstrate the potential of blockchain–cloud integration in healthcare, real-world implementation poses significant challenges. Legacy system integration, change management, and operational complexities present hurdles that must be overcome for successful deployment. Addressing these challenges requires careful planning, collaboration, and a commitment to overcoming obstacles to unlock the transformative potential of blockchain and cloud technologies in healthcare.

- **Real-World Implementation Challenges:** While theoretical models and simulations show promise, this review highlights the significant gap between conceptual designs and practical, large-scale implementation in real-world healthcare environments. Addressing issues like legacy system integration, change management, and operational complexities is crucial for successful deployment.

### *5.2. Recommendations for Future Research*

To further the progress in integrating blockchain and cloud computing for healthcare data management, this section offers detailed recommendations for future research. These recommendations focus on developing scalable consensus mechanisms, interoperability standards, quantum-resistant cryptographic solutions, cost-efficient hybrid architectures, regulatory compliance frameworks, and sustainable business models. By addressing these areas, future research can contribute to the practical realization of secure, decentralized, and high-performing systems for healthcare data management.

### 5.2.1. Implementation and Integration Challenges

This section explores the various challenges associated with implementing and integrating blockchain and cloud technologies in healthcare settings. We will delve into the practical difficulties such as integrating with legacy systems, managing organizational change, and handling operational complexities. Additionally, we will examine strategies for collaborating with IT professionals and healthcare staff to develop effective integration and training solutions, ensuring successful deployment.

- **Real-World Implementation Challenges:** This review highlights the significant gap between conceptual designs and practical, large-scale implementation in real-world healthcare environments. Addressing issues such as legacy system integration, change management, and operational complexities is crucial for successful deployment. Collaboration with IT professionals and healthcare staff is recommended to develop appropriate integration and training strategies.
- **Usability and Adoption Strategies:** Future studies should investigate strategies to improve the usability and adoption of blockchain–cloud solutions in healthcare, such as user-friendly interfaces, change management processes, and training programs for healthcare professionals. Research should focus on understanding the needs and challenges of end-users, and developing solutions that are intuitive and easy to implement. Additionally, case studies and pilot projects should be considered to evaluate and refine these strategies in real-world environments.

### 5.2.2. Technological and Research Advancements

In this section, we will discuss the technological advancements and research directions necessary to enhance blockchain–cloud integration in healthcare. Topics include the development of scalable consensus mechanisms, quantum-resistant cryptographic solutions, and cost-efficient hybrid architectures. We will also address the need for robust regulatory compliance frameworks and sustainable models to support the long-term viability of these technologies.

- **Exploration of Scalable Consensus Mechanisms:** Future research should prioritize the development of novel consensus algorithms specifically designed for blockchain–cloud integration in healthcare settings. These algorithms should aim to improve scalability while maintaining robust security measures. Investigating consensus mechanisms that can efficiently handle the increasing transaction volumes associated with healthcare data management is crucial. Emphasis should be placed on mechanisms that can adapt to dynamic network conditions and evolving technological landscapes, ensuring long-term scalability and performance.

- **Quantum-Resistant Solutions Research:** Given the looming threat of quantum computing, future research endeavors should prioritize the research and development of cryptographic techniques resistant to quantum attacks. This is particularly crucial for ensuring the long-term security and resilience of blockchain–cloud systems in healthcare. Investigating quantum-resistant encryption algorithms, digital signature schemes, and authentication mechanisms will be instrumental in future-proofing these systems against emerging threats posed by advancements in quantum computing. Collaborations with academic and research institutions in cryptography are recommended.

- **Technological Advancements and Innovative Models:** The reviewed literature highlights promising technological advancements and innovative models aimed at overcoming existing hurdles. Hybrid on-chain/off-chain architectures, advances in cryptography and trusted execution environments, and the emergence of decentralized storage networks represent significant strides in addressing scalability, efficiency, and security concerns.

- **Development of Interoperability Standards:** To enhance seamless data exchange and accessibility between blockchain and cloud systems, future research should focus on developing standardized protocols and interoperability standards. These standards will play a pivotal role in enabling different platforms and systems to communicate effectively, facilitating secure and efficient data transfer. Collaboration with international standardization organizations is suggested to ensure global acceptance.

### 5.2.3. Sustainability and Compliance

This section focuses on the sustainability and compliance aspects of blockchain–cloud integration in healthcare. We will explore the development of cost-efficient hybrid architectures, regulatory compliance frameworks, and long-term sustainability models. The goal is to ensure that these technologies are not only effective and secure but also economical via **Cost-Efficient Hybrid Architectures Exploration:** Efforts should be directed toward exploring cost-effective hybrid architectures that optimize resource utilization in blockchain–cloud environments without compromising security and privacy. Research in this area should focus on designing efficient resource allocation models, minimizing overhead costs, and enhancing scalability and performance. These hybrid architectures should strike a balance between centralized control for governance and decentralized structures for data integrity and transparency, catering to the unique requirements of healthcare data management.

- **Regulatory Compliance Frameworks Design:** Future research should address the complex legal and compliance challenges associated with blockchain–cloud integration in healthcare by designing robust regulatory compliance frameworks. These frameworks should ensure adherence to data protection regulations, privacy laws,

and healthcare standards while promoting innovation and technological advancement. Developing clear guidelines and protocols for data sharing, consent management, and regulatory reporting will foster trust and confidence in blockchain–cloud infrastructures within the healthcare sector.

- **Long-Term Sustainability Models Proposition:** Research efforts should focus on proposing sustainable business models for maintaining blockchain–cloud infrastructures in healthcare settings. Considerations should include factors such as resource consumption, environmental impact, cost-effectiveness, and scalability. Designing sustainable models that align with economic, social, and environmental goals will be crucial for the long-term viability and adoption of blockchain–cloud integration in healthcare. Collaboration with economists and sustainability experts is recommended to develop these proposals.

The integration of blockchain and cloud computing in healthcare data management holds significant promise. Key findings indicate that blockchain enhances data security and privacy through its decentralized, immutable nature, and improves data accessibility and interoperability when combined with cloud computing.

However, significant challenges such as scalability, performance, and regulatory compliance need to be addressed. Technological advancements like hybrid architectures and quantum-resistant cryptographic solutions offer potential solutions.

Future research should focus on developing cost-efficient, sustainable models and standardized protocols to ensure long-term viability. Multidisciplinary collaboration among blockchain experts, healthcare professionals, policymakers, and cloud providers are essential to overcome these challenges and fully realize the benefits of blockchain–cloud integration in healthcare.

**Author Contributions:** This review study has the following individual contributions: Conceptualization, L.J.R.L. and D.M.M.; methodology, L.J.R.L. and L.H.M.P.; software, D.M.M., L.H.M.P., and A.F.C.A.; validation, L.J.R.L. and W.R.R.; formal analysis, L.J.R.L. and D.M.M.; research, L.J.R.L., D.M.M., L.H.M.P., A.F.C.A., and W.R.R.; resources, L.J.R.L., D.M.M., L.H.M.P., A.F.C.A., and W.R.R.; data curation, L.J.R.L., D.M.M., and L.H.M.P.; writing—original draft preparation, D.M.M., and L.H.M.P.; writing—review and editing, L.J.R.L. and D.M.M.; visualization, L.J.R.L. and L.H.M.P.; supervision, L.J.R.L.; project management, L.J.R.L. and W.R.R.; acquisition financing, L.J.R.L., D.M.M., L.H.M.P., A.F.C.A., and W.R.R. All authors have read and agreed to the published version of the manuscript.

**Funding:** This research received no external funding.

**Data Availability Statement:** No new data were created.

**Acknowledgments:** The authors acknowledge Universidad El Bosque for support to research project code AUX-2023-02-01, and access to scientific databases.

**Conflicts of Interest:** The authors declare no conflicts of interest.

## Appendix A

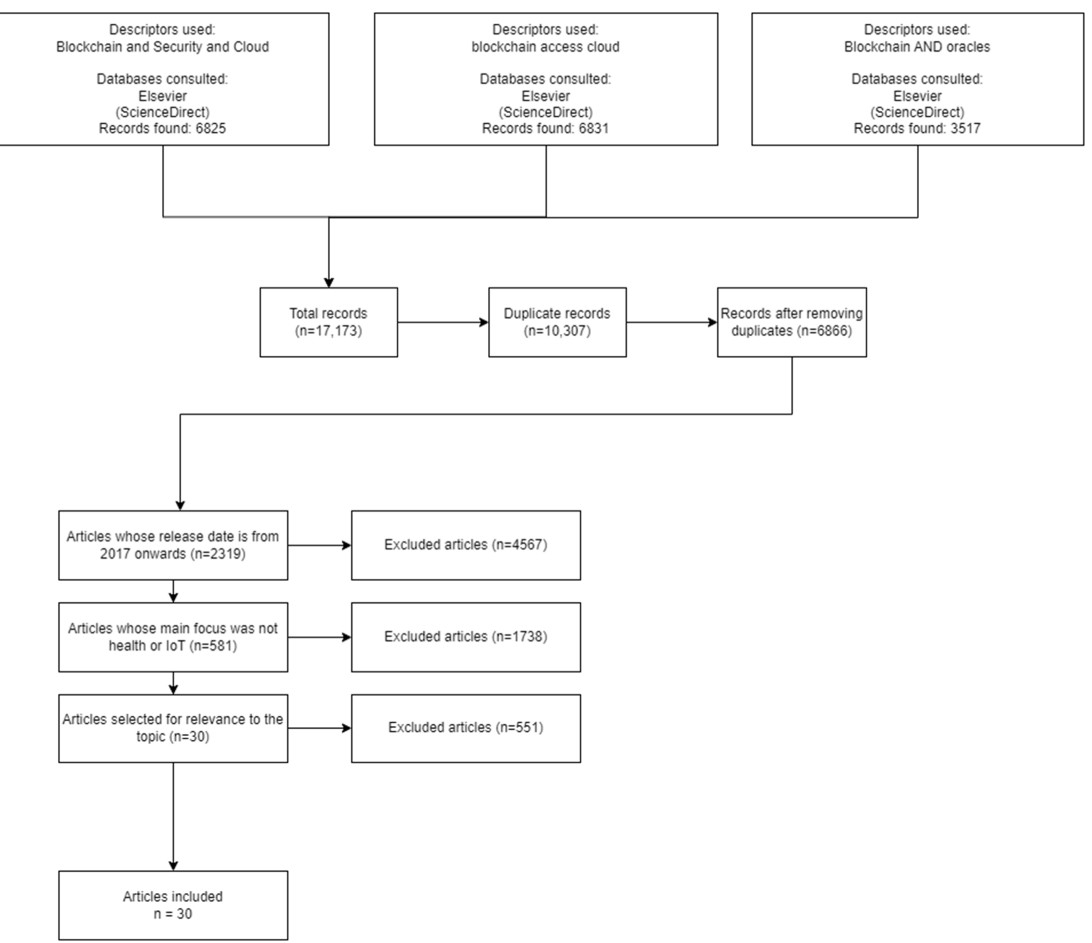

**Figure A1.** Flowchart for selected papers on Elsevier (ScienceDirect).

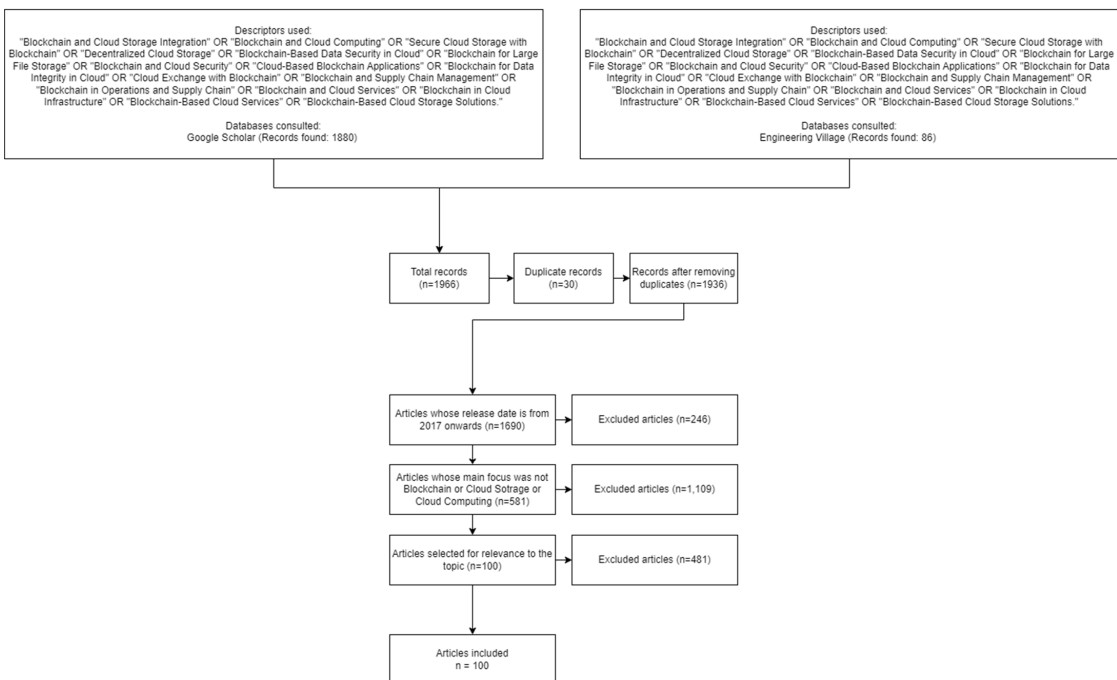

**Figure A2.** Flowchart for selected papers on Google Scholar and Engineering Village.

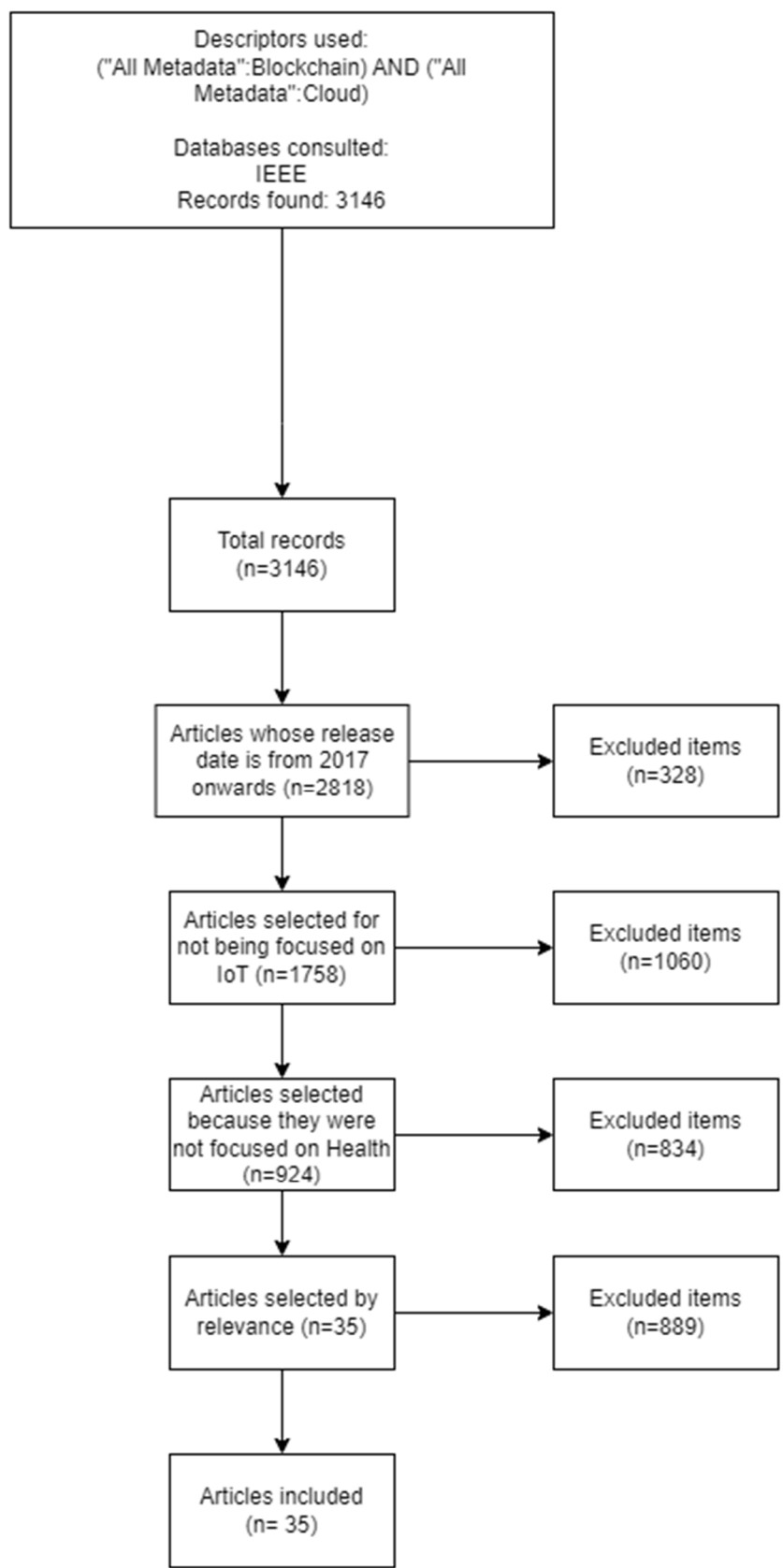

**Figure A3.** Flowchart for selected papers on IEEE.

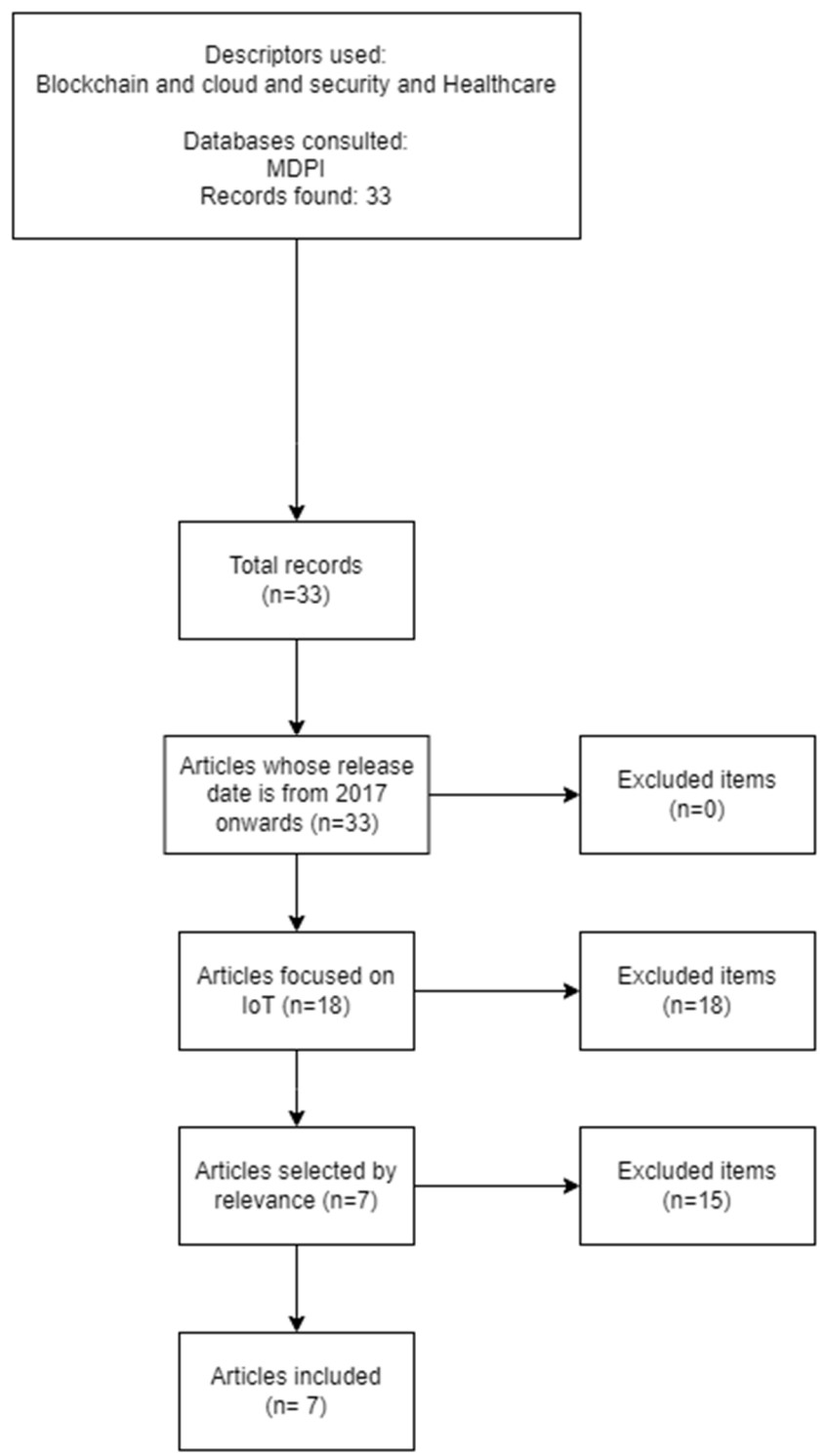

**Figure A4.** Flowchart for selected papers on MDPI.

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
