# Peer review of "Hybrid Architectures Used in the Protection of Large Healthcare Records Based on Cloud and Blockchain Integration: A Review"

_computers, doi:10.3390/computers13060152_

Round 1
Reviewer 1 Report
Comments and Suggestions for Authors
The authors follow well the PRISMA guideline. Still, review papers should "critically analyze the strengths and weaknesses of existing studies, offer a foundation for new research, and trace the evolution of ideas and methodologies". This part is a bit lackluster. The key findings and the recommandations for future research from Conclusions seem to be overly general. We don't really need comprehensive reviews to know that "pressing challenges that still need to be addressed, including efficiency, scalability, costs, interoperability, quantum computing resistance, and balancing centralization with decentralization. Overcoming these limitations is crucial for fully realizing the potential of blockchain-cloud integration."
Comments on the Quality of English LanguageNo big isssues. Some sentences sound a bit strange.
Author Response
Dear Reviewer 1
Computers MDPI Journal
Thank you very much for reviewing our manuscript. We also greatly appreciate the reviewer for their complimentary comments and suggestions. Newly, we have carried out the adjust that the reviewers suggested and revised the manuscript accordingly.
We revised the manuscript and we propounded a lot of changes have taken place. Please find attached to the point-by-point response to reviewer's concerns. We hope that you find our responses satisfactory and that the manuscript is now acceptable for publication. So, we have sent the revised manuscript, and the version containing all the changes to be visible.
Sincerely,
Leonardo Juan Ramirez Lopez on behalf of the authors
Reviewer 1
Newly, we revised our manuscript, and propounded a lot of changes have taken place. So, we have sent the revised manuscript, and the version containing all the changes to be visible.
At the following, the points mentioned by the reviewers will be discussed:
Comment 1: The authors follow well the PRISMA guideline. Still, review papers should "critically analyze the strengths and weaknesses of existing studies, offer a foundation for new research, and trace the evolution of ideas and methodologies". This part is a bit lackluster.
Response 1: Thank you very much for your valuable comment. The study employs a scoping review to comprehensively examine and map the existing literature on the topic. This approach allows for a broad exploration of the research landscape, providing a comprehensive overview of relevant studies and their key findings. This is why a critical analysis is carried out on each article found, always seeking to reach innovative results that allow us to answer each of the questions posed. In our study, not only the title and summary are reviewed, but also the methodology, results, discussion of results and the conclusions provided.
We value your important comment and we have included 16 detailed explanations to the response to RQ6: How are Large Healthcare Files Currently Secured in Cloud-Blockchain Integration, and What Methods and Techniques are currently being Utilized?
In six new sections from 3.3.1 to 3.3.6, it includes five new figures.
Location: from line 414 to 769.
Comment 2: The key findings and the recommendations for future research from Conclusions seem to be overly general.
Response 2: Your comment is of high value to raise the quality of the manuscript. We have expanded the conclusions with the section “5.1 Key findings”, and the section “5.2 Recommendations for Future Research”. With these critical analyzes readers will be clear about where future research related to this topic and its technologies are going.
Location: from line 1096 to 1220.
Comment 3: We don't really need comprehensive reviews to know that "pressing challenges that still need to be addressed, including efficiency, scalability, costs, interoperability, quantum computing resistance, and balancing centralization with decentralization. Overcoming these limitations is crucial for fully realizing the potential of blockchain-cloud integration
Response 3: We completely agree with reviewer 1, and this is why we included the new “section 3.3.5 Oracles: A reinforcement learning model for the reliability of blockchain oracles”. Furthermore, in “section 4.8 Technological Landscape for Secure Cloud-Blockchain Integration in Healthcare”, we support our argument with four important related studies.
Location: from line 662 to 731, and from line 1069 to 1070.
Comment 4: No big issues. Some sentences sound a bit strange.
Response 4: Thank you very much for your valuable comment. We have completely revised the English edition of the manuscript.
The authors appreciate all the comments that strengthen the quality of the article.
The authors

Reviewer 2 Report
Comments and Suggestions for Authors
This paper provides a review on the existing approaches integrating blockchain and cloud solutions for the protection of healthcare data. The authors relied on the PRISMA methodology in selecting the research papers. One of the criteria was to select papers published between 2017 and 2023, to focus just on modern and not out-of-date technologies. In Section 1 the authours report the research questions to which they reply in Section 3. Their findings are then discussed in Section 4, highlighting future research directions.
The paper is well-written, with acceptable coverage of research papers, and the review they provide seems to consider all the important aspects of the problem they discuss.
Minor comments/typo
- line 165: missing subject
- line 343: table 6 -> Table 6
- line 348: RQ6 in \textbb
Author Response
Dear Reviewer 2
Computers MDPI Journal
Thank you very much for reviewing our manuscript. We also greatly appreciate the reviewer for their complimentary comments and suggestions. Newly, we have carried out the adjust that the reviewers suggested and revised the manuscript accordingly.
We revised the manuscript and we propounded a lot of changes have taken place. Please find attached to the point-by-point response to reviewer's concerns. We hope that you find our responses satisfactory and that the manuscript is now acceptable for publication. So, we have sent the revised manuscript, and the version containing all the changes to be visible.
Sincerely,
Leonardo Juan Ramirez Lopez on behalf of the authors
Reviewer 2
Newly, we revised our manuscript, and propounded a lot of changes have taken place. So, we have sent the revised manuscript, and the version containing all the changes to be visible.
At the following, the points mentioned by the reviewers will be discussed:
Comment 1: This paper provides a review on the existing approaches integrating blockchain and cloud solutions for the protection of healthcare data. The authors relied on the PRISMA methodology in selecting the research papers. One of the criteria was to select papers published between 2017 and 2023, to focus just on modern and not out-of-date technologies. In Section 1 the authors report the research questions to which they reply in Section 3. Their findings are then discussed in Section 4, highlighting future research directions.
The paper is well-written, with acceptable coverage of research papers, and the review they provide seems to consider all the important aspects of the problem they discuss.
Response 1: We highly appreciate this comment because it encourages us to continue our research with the best of spirits.
Comment 2: Minor comments/typo: - line 165: missing subject. - line 343: table 6 -> Table 6. - line 348: RQ6 in \textbb
Response 2: Thank you for your comment, the authors had reviewed the paper. We correct what reviewer 2 requests.
Location: figure 6: line 557; RQ6: line 371; and others
The authors appreciate all the comments that strengthen the quality of the article.
The authors

Reviewer 3 Report
Comments and Suggestions for Authors
The paper provides a comprehensive overview of the integration of blockchain and cloud computing in healthcare data management, addressing key challenges and potential benefits. Here are some comments for the authors to consider :
- Abbreviations should be written in full the first time that they are used in the paper;
- The paper mentions the use of the PRISMA methodology for study selection, which is suitable for systematic reviews and meta-analyses. However, it would be helpful to provide more detail on how the PRISMA methodology was applied, including specific inclusion and exclusion criteria, search strategies, and data extraction processes ;
- The literature review provides a comprehensive overview of existing research on blockchain, cloud computing, and their integration in healthcare. However, it would be beneficial to include more critical analysis and synthesis of the literature to identify key themes, gaps, and areas for further investigation.
Author Response
Dear Reviewer 3
Computers MDPI Journal
Thank you very much for reviewing our manuscript. We also greatly appreciate the reviewer for their complimentary comments and suggestions. Newly, we have carried out the adjust that the reviewers suggested and revised the manuscript accordingly.
We revised the manuscript and we propounded a lot of changes have taken place. Please find attached to the point-by-point response to reviewer's concerns. We hope that you find our responses satisfactory and that the manuscript is now acceptable for publication. So, we have sent the revised manuscript, and the version containing all the changes to be visible.
Sincerely,
Leonardo Juan Ramirez Lopez on behalf of the authors
Reviewer 3
Newly, we revised our manuscript, and propounded a lot of changes have taken place. So, we have sent the revised manuscript, and the version containing all the changes to be visible.
At the following, the points mentioned by the reviewers will be discussed:
Comment 1: The paper provides a comprehensive overview of the integration of blockchain and cloud computing in healthcare data management, addressing key challenges and potential benefits. Here are some comments for the authors to consider: Abbreviations should be written in full the first time that they are used in the paper.
Response 1: We kindly appreciate this comment and made a complete review of the abbreviations, where they are fully explained the first time they are used in the article.
Location: lines 13; 93 to 94; 164 to 166; 415; 436; 446; 506; 716.
Comment 2: - The paper mentions the use of the PRISMA methodology for study selection, which is suitable for systematic reviews and meta-analyses. However, it would be helpful to provide more detail on how the PRISMA methodology was applied, including specific inclusion and exclusion criteria, search strategies, and data extraction processes.
Response 2: Thank you very much for your valuable comment. The study employs a scoping review to comprehensively examine and map the existing literature on the topic. The articles were selected using the PRISMA methodology and subtracted from 4 different databases such as: Elsevier (ScienceDirect), Google Scholar, IEEE Xplore and MDPI. Additionally, the PRSMA makes it easier to understand what needs to be done at each stage of the review. Location: line 241 to 250; and 167 to 187
We added several new sections: 2.2 Statistics of the use of PRISMA in scientific review type articles; 2.3 Importance of PRISMA; 2.4 Search engines and search equations; and 2.5 Incorporation and Exclusion Parameters.
Location: line190 to 263
Comment 3: The literature review provides a comprehensive overview of existing research on blockchain, cloud computing, and their integration in healthcare. However, it would be beneficial to include more critical analysis and synthesis of the literature to identify key themes, gaps, and areas for further investigation.
Response 3: Thank you very much for your valuable contribution of knowledge. This is why we have included a new section 3.3 Innovative Research Perspectives, with several items: 3.3.1 Encryption; 3.3.2 Access control; 3.3.3 Off chain; 3.3.4 Cryptographic techniques; 3.3.5 Oracles; and 3.3.6 Secure Data Sharing and Consent Management. Likewise, 5 new figures for the reader's better understanding.
Location: line 405 to 769
We have expanded the conclusions with the section “5.1 Key findings”, and the section “5.2 Recommendations for Future Research”. With these critical analyzes readers will be clear about where future research related to this topic and its technologies are going.
Location: line 1096 to 1222.
The authors appreciate all the comments that strengthen the quality of the article.
The authors

Reviewer 4 Report
Comments and Suggestions for Authors
The topics covered by the manuscript are interesting in themselves and still attract the attention of a very large audience. The language and editing used are quite good, even if they require rereading and revision (for example it would be good to make the acronym PRISMA explicit the first time it appears both in the abstract and in the body of the manuscript; remove the extra period in the line 45 on page 2; etc.). The weakest point of the work, in my opinion, is the contents: for the most part it is reduced to a quantitative-statistical study on the literature, with a marked attention to the abstracts. A work of this kind should instead be a reasoned review written after in-depth study and profound knowledge of the works being discussed. It is not very appropriate to entrust the most important role in the selection of articles to search engines. This way important articles that do not contain all the keywords will not be considered at all. Thus, the most innovative and general works, applicable in different contexts, are penalized (for example more technical papers such as doi 10.3390/math10173040, or 10.3390/s21217300, to stay with MDPI, or many others). There is also a shortage of comparisons, critical approaches and insights. On the contrary, however, the text is sometimes a bit verbose and repetitive, which can also be deduced from the unusual length of the abstract.
Comments on the Quality of English LanguageThe English language is quite good.
Author Response
Dear Reviewer 4
Computers MDPI Journal
Thank you very much for reviewing our manuscript. We also greatly appreciate the reviewer for their complimentary comments and suggestions. Newly, we have carried out the adjust that the reviewers suggested and revised the manuscript accordingly.
We revised the manuscript and we propounded a lot of changes have taken place. Please find attached to the point-by-point response to reviewer's concerns. We hope that you find our responses satisfactory and that the manuscript is now acceptable for publication. So, we have sent the revised manuscript, and the version containing all the changes to be visible.
Sincerely,
Leonardo Juan Ramirez Lopez on behalf of the authors
Reviewer 4
Newly, we revised our manuscript, and propounded a lot of changes have taken place. So, we have sent the revised manuscript, and the version containing all the changes to be visible.
At the following, the points mentioned by the reviewers will be discussed:
Comment 1: The topics covered by the manuscript are interesting in themselves and still attract the attention of a very large audience.
Response 1: We highly appreciate this comment because it encourages us to continue our research with the best of spirits.
Comment 2: The language and editing used are quite good, even if they require rereading and revision (for example it would be good to make the acronym PRISMA explicit the first time it appears both in the abstract and in the body of the manuscript; remove the extra period in the line 45 on page 2; etc.).
Response 2: We kindly appreciate this comment and made a complete review of the abbreviations, where they are fully explained the first time they are used in the article.
Location: lines 13; 93 to 94; 164 to 166; 415; 436; 446; 506; 716.
Comment 3: The weakest point of the work, in my opinion, is the contents: for the most part it is reduced to a quantitative-statistical study on the literature, with a marked attention to the abstracts.
Response 3: Thank you very much for your important comment. The study employs a scoping review to comprehensively examine and map the existing literature on the topic. This approach allows for a broad exploration of the research landscape, providing a comprehensive overview of relevant studies and their key findings. This is why a critical analysis is carried out on each article found, always seeking to reach innovative results that allow us to answer each of the questions posed. In our study, not only the title and summary are reviewed, but also the methodology, results, discussion of results and the conclusions provided.
We value your important comment and we have included 16 detailed explanations to the response to RQ6: How are Large Healthcare Files Currently Secured in Cloud-Blockchain Integration, and What Methods and Techniques are currently being Utilized?
In six new sections from 3.3.1 to 3.3.6, it includes five new figures.
Location: from line 414 to 769.
We added several new sections: 2.2 Statistics of the use of PRISMA in scientific review type articles; 2.3 Importance of PRISMA; 2.4 Search engines and search equations; and 2.5 Incorporation and Exclusion Parameters.
Location: line 191 to 222
Comment 4: A work of this kind should instead be a reasoned review written after in-depth study and profound knowledge of the works being discussed. It is not very appropriate to entrust the most important role in the selection of articles to search engines. This way important articles that do not contain all the keywords will not be considered at all. Thus, the most innovative and general works, applicable in different contexts, are penalized (for example more technical papers such as doi 10.3390/math10173040, or 10.3390/s21217300, to stay with MDPI, or many others). There is also a shortage of comparisons, critical approaches and insights.
Response 4: Thank you very much for your valuable contribution of knowledge. This is why we have included a new section 3.3 Innovative Research Perspectives, with several items: 3.3.1 Encryption; 3.3.2 Access control; 3.3.3 Off chain; 3.3.4 Cryptographic techniques; 3.3.5 Oracles; and 3.3.6 Secure Data Sharing and Consent Management. Likewise, 5 new figures for the reader's better understanding.
Location: line 405 to 769.
In this same sense, the authors appreciate the recommendation to include these important technical papers:
doi 10.3390/math10173040, is included as reference 109. Line 1574
doi 10.3390/s21217300, is included as reference 110. Line 1576
Comment 5: On the contrary, however, the text is sometimes a bit verbose and repetitive, which can also be deduced from the unusual length of the abstract.
Response 5: We kindly acknowledge this feedback and have conducted a full review of the summary and editing language.
The abstract was edited to 19 words.
Location: from line 9 to 22
The authors appreciate all the comments that strengthen the quality of the article.
The authors

Round 2
Reviewer 1 Report
Comments and Suggestions for Authors
In my opinion the article still caries many of the previous problems. The key findings and the recommendations for future research from Conclusions are still overly general but now in more words. Many paragraphs still don't sound natural to me. For example:
• Development of Interoperability Standards: To enhance seamless data exchange and accessibility between blockchain and cloud systems, future research should focus on developing standardized protocols and interoperability standards. These standards will play a pivotal role in enabling different platforms and systems to communicate effectively, facilitating secure and efficient data transfer. Emphasizing compatibility and interoperability across diverse blockchain implementations and cloud environments is essential for promoting widespread adoption and integration in healthcare infrastructures.
• Quantum-Resistant Solutions Research: Given the looming threat of quantum computing, future research endeavors should prioritize the research and development of cryptographic techniques resistant to quantum attacks. This is particularly crucial for ensuring the long-term security and resilience of blockchain-cloud systems in healthcare. Investigating quantum-resistant encryption algorithms, digital signature
Author Response
Dear Reviewer 1
Computers MDPI Journal
Ref: Round two
Thank you very much for reviewing our manuscript. We also greatly appreciate the reviewer for their complimentary comments and suggestions. Newly, we have carried out the adjust that the reviewers suggested and revised the manuscript accordingly.
We revised the manuscript and we propounded a lot of changes have taken place. Please find attached to the point-by-point response to reviewer's concerns. We hope that you find our responses satisfactory and that the manuscript is now acceptable for publication. So, we have sent the revised manuscript, and the version containing all the changes to be visible.
Sincerely,
Leonardo Juan Ramirez Lopez on behalf of the authors
Reviewer 1
Newly, we revised our manuscript, and propounded a lot of changes have taken place. So, we have sent the revised manuscript, and the version containing all the changes to be visible.
At the following, the points mentioned by the reviewers will be discussed:
Comment 1: In my opinion the article still caries many of the previous problems. The key findings and the recommendations for future research from Conclusions are still overly general but now in more words. Many paragraphs still don't sound natural to me. For example:
- Development of Interoperability Standards: To enhance seamless data exchange and accessibility between blockchain and cloud systems, future research should focus on developing standardized protocols and interoperability standards. These standards will play a pivotal role in enabling different platforms and systems to communicate effectively, facilitating secure and efficient data transfer. Emphasizing compatibility and interoperability across diverse blockchain implementations and cloud environments is essential for promoting widespread adoption and integration in healthcare infrastructures.
- Quantum-Resistant Solutions Research: Given the looming threat of quantum computing, future research endeavors should prioritize the research and development of cryptographic techniques resistant to quantum attacks. This is particularly crucial for ensuring the long-term security and resilience of blockchain-cloud systems in healthcare. Investigating quantum-resistant encryption algorithms, digital signature.
Response 1: We greatly appreciate this comment because it encouraged us to review, correct and write recommendations for future research and rethink the most natural and forceful conclusions.
Location in the manuscript: Since line 1074 until line 1311.
Please see the attachment.
The authors appreciate all the comments that strengthen the quality of the article.
Reviewer 4 Report
Comments and Suggestions for Authors
Already from the first version of the manuscript, a format that did not conform to the MDPI one for the bibliography was noted, and it is also unchanged in the new version.
For example, the first item reads:
[1] S. Bhari and S. J. Quraishi, “Blockchain and Cloud Computing-A Review,” in 2022 International 604 Conference on Machine Learning, Big Data, Cloud and Parallel Computing, COM-IT-CON 2022, 2022. 605 doi: 10.1109/COM-IT-CON54601.2022.9850499.
With the MDPI format you should have something similar to this:
[1] Bhari, S.; Quraishi, S.J. Blockchain and Cloud Computing-A Review. In ...
In particular, the comma at the end of article titles never goes inside " ". In reference [109] there also appears a strange symbol that must be removed, etc.
It is therefore recommended to review the entire bibliography, and this can be done very easily by adopting a BibTeX file.
Author Response
Dear Reviewer 4
Computers MDPI Journal
Ref: Round two
Thank you very much for reviewing our manuscript. We also greatly appreciate the reviewer for their complimentary comments and suggestions. Newly, we have carried out the adjust that the reviewers suggested and revised the manuscript accordingly.
We revised the manuscript and we propounded a lot of changes have taken place. Please find attached to the point-by-point response to reviewer's concerns. We hope that you find our responses satisfactory and that the manuscript is now acceptable for publication. So, we have sent the revised manuscript, and the version containing all the changes to be visible.
Sincerely,
Leonardo Juan Ramirez Lopez on behalf of the authors
Reviewer 4
Newly, we revised our manuscript, and propounded a lot of changes have taken place. So, we have sent the revised manuscript, and the version containing all the changes to be visible.
At the following, the points mentioned by the reviewers will be discussed:
Comment 1: Already from the first version of the manuscript, a format that did not conform to the MDPI one for the bibliography was noted, and it is also unchanged in the new version.
For example, the first item reads:
[1] S. Bhari and S. J. Quraishi, “Blockchain and Cloud Computing-A Review,” in 2022 International 604 Conference on Machine Learning, Big Data, Cloud and Parallel Computing, COM-IT-CON 2022, 2022. 605 doi: 10.1109/COM-IT-CON54601.2022.9850499.
With the MDPI format you should have something similar to this:
[1] Bhari, S.; Quraishi, S.J. Blockchain and Cloud Computing-A Review. In ...
In particular, the comma at the end of article titles never goes inside " ". In reference [109] there also appears a strange symbol that must be removed, etc.
It is therefore recommended to review the entire bibliography, and this can be done very easily by adopting a BibTeX file.
Response 1: We greatly appreciate this comment because we have completely reviewed the references of the manuscript and they conformed to the MDPI format.
Location in the manuscript: since line 1379 until line 1702
Please see the attachment
The authors appreciate all the comments that strengthen the quality of the article.